# The fate of early perichondrial cells in developing bones

Yuki Matsushita[1,2], Angel Ka Yan Chu[3], Chiaki Tsutsumi-Arai[1], Shion Orikasa[1], Mizuki Nagata[1], Sunny Y. Wong[4], Joshua D. Welch[3], Wanida Ono[1] & Noriaki Ono [1] ✉

In endochondral bone development, bone-forming osteoblasts and bone marrow stromal cells have dual origins in the fetal cartilage and its surrounding perichondrium. However, how early perichondrial cells distinctively contribute to developing bones remain unidentified. Here we show using in vivo cell-lineage analyses that Dlx5+ fetal perichondrial cells marked by *Dlx5-creER* do not generate cartilage but sustainably contribute to cortical bone and marrow stromal compartments in a manner complementary to fetal chondrocyte derivatives under the regulation of Hedgehog signaling. Postnatally, Dlx5+ fetal perichondrial cell derivatives preferentially populate the diaphyseal marrow stroma with a dormant adipocyte-biased state and are refractory to parathyroid hormone-induced bone anabolism. Therefore, early perichondrial cells of the fetal cartilage are destined to become an adipogenic subset of stromal cells in postnatal diaphyseal bone marrow, supporting the theory that the adult bone marrow stromal compartments are developmentally prescribed within the two distinct cells-of-origins of the fetal bone anlage.

Bones are incredibly malleable organs capable of growing exponentially in early life and maintaining their structures and functions throughout life. The inherent flexibility of bones is endowed at least in part by the unique property of bone-forming osteoblasts that can be generated from multiple cellular origins[1–6]. Most bones in mammals are formed through the endochondral pathway, in which initial cartilage templates established in a vasculature-free environment are gradually replaced by bones associated with invading vasculature[7]. The two most developmentally important cellular sources of endochondral bones lie in the cartilage template proper and its surrounding perichondrium. It is now commonly accepted that fetal chondrocytes within cartilage templates can transform into osteoblasts and marrow stromal cells to establish the nascent marrow space[2,5,8–10]. Cells in the perichondrium are equally important, as these cells can contribute both to the cortical bone and the marrow space[11]. The partnership between the cartilage template and the perichondrium provides the anatomical foundation of

endochondral bones, ultimately establishing diverse compartments in postnatal long bones.

The perichondrium is composed of multiple layers of elongated fibroblasts in a vasculature-rich environment. Cells constituting the perichondrium of the fetal cartilage are not completely defined. The osteogenic perichondrium is the inner layer of the perichondrium where the first osteoblasts appear in endochondral bones[12]. The formation of osteoblast precursor cells in the osteogenic perichondrium is executed at least in part due to the actions of Indian Hedgehog (Ihh) released from adjacent pre-hypertrophic chondrocytes[13]. Osterix (Osx)-expressing cells are located in the innermost layer of the perichondrium and represent the committed precursor cells of the osteoblast lineage. Osx+ cells translocate into the nascent marrow space, and their descendants eventually populate the area as osteoblasts and marrow stromal cells[4]. Importantly, these Osx+ cell derivatives are transient, and eventually disappear from the perichondrium and the marrow space in subsequent postnatal stages. The identity of

[1]University of Texas Health Science Center at Houston School of Dentistry, Houston, TX 77054, USA. [2]Department of Cell Biology, Nagasaki University Graduate School of Biomedical Sciences, Nagasaki 852-8588, Japan. [3]Department of Computational Medicine and Bioinformatics, University of Michigan, Ann Arbor, MI 48109, USA. [4]Department of Dermatology, University of Michigan Medical School, Ann Arbor, MI 48109, USA. ✉e-mail: noriaki.ono@uth.tmc.edu

earlier perichondrial cells that constantly replace Osx[+] cells in the perichondrium remain unidentified.

Previous studies reported several constitutively active *cre* and inducible *creER* lines that can target perichondrial cells, including *Prrx1-creER*[14], *Hoxa11-creER*[15,16], *Sox9-creER*[17], *Axin2-creER*[18], *Ctsk-cre*[19], and *Fsp1-cre*[20,21]. While these studies substantially contributed to the field, these reported lines may have potential limitations. In particular, these *creER* lines also mark at least some of chondrocytes within the growth plate. Moreover, *Hoxa11-creER* is active exclusively in the zeugopod bones (ulna, radius, tibia, and fibula)[15], while *Sox9-creER* is specific to perichondrial cells exclusively in the rib cartilage[17]. Additionally, the constitutively active *cre* lines lack temporal control and have activities in other cell types, such as osteoclasts (*Ctsk-cre*) and other fibroblasts (*Fsp1-cre*). Therefore, a perichondrial cell-specific inducible mouse genetic tool would facilitate the understanding of the function of early perichondrial cells.

In this study, we aimed to achieve two major goals. First, we sought to define the identity of as-yet unidentified early perichondrial cells that can provide a perpetual source of osteoblasts and marrow stromal cells, using a combination of single-cell RNA-seq analyses and in vivo lineage-tracing approaches. Second, we sought to define how these identified early perichondrial cells contribute postnatally to functionally distinct compartments of developing bones, including the cortical bone and the bone marrow, and to determine whether osteoblasts and bone marrow stromal cells derived from the two cells-of-origins have distinct features and functions, using comparative in vivo lineage-tracing approaches. The cell type-specific *creER* lines – *Dlx5-creER* for early fetal perichondrial cells and *Fgfr3-creER* for fetal chondrocytes – highlight distinctive cell fates and contributions of the two cell types. Our findings support the theory that the adult bone marrow stromal compartments are developmentally prescribed within the two distinct cells-of-origins of the fetal cartilage.

## Results

### Putative cells-of-origins of the fetal cartilage

First, to discover the putative cellular origins of the fetal cartilage, we performed single-cell RNA-sequencing (scRNA-seq) analyses of the chondrocyte-perichondrial cell lineage at embryonic day (E) 13.5 in mice. Both perichondrial cells and chondrocytes at the fetal stage can be fate-mapped by a *cre* recombinase driven by a *Col2a1* promoter (*Col2a1-cre*)[22] (Fig. 1a). We dissociated limb mesenchymal cells of *Col2a1-cre*; *R26R*[tdTomato] mice at E13.5, and profiled fluorescence-activated cell sorting (FACS)-isolated Col2a1[cre]-tdTomato[+] cells using the 10X Chromium Single-Cell Gene Expression Solution platform (Fig. 1b, c and Supplementary Fig. 1a). Louvain community detection after cell cycle regression revealed 9 clusters (Fig. 1d), including three clusters of chondrocytes abundant in *Sox9* (Cluster 0,1,2; Fig. 1e and Supplementary Fig. 1b, see also Fig. 1a for immunostaining) and two clusters of perichondrial cells abundant in *Prrx1* (Cluster 4,5).

The fetal chondrocyte-perichondrial cell lineage was largely contiguous and shared an overlapping set of marker genes. Of the chondrocyte clusters, cells in Cluster 1 were enriched for *Col2a1*, *Fgfr3*, *Acan*, *Ihh*, and *Sp7*, representing relatively mature chondrocytes (Fig. 1e). Among the perichondrial cell clusters, cells in Cluster 5 were enriched for *Runx2* and *Sp7*, representing perichondrial cells partially committed to the osteoblast lineage[4,23]. Both *Dlx5* and *Prrx1* were enriched in Cluster 4 and 5. *Dlx5* is an upstream regulator of Runx2[24] and a co-factor of Sp7 (also known as osterix, Osx)[25]. *Dlx5* was enriched both in Cluster 4 and 5, identified as the most significantly differentially expressed gene in Cluster 5 and most abundantly expressed at the border of Cluster 4 and 5 (Fig. 1e). In contrast, *Prrx1* was diffusely expressed in other clusters including chondrocytes (Fig. 1e), indicating *Dlx5* might serve as a reliable cell type-specific marker for early perichondrial cells.

We further examined in vivo expression patterns of potential cell type-specific genes that we identified above using independent approaches, including *Fgfr3-GFP; Sp7(Osx)-mCherry* double transgenic reporters, RNAscope assays and SOX9 immunostaining. Importantly, SOX9 proteins and Fgfr3-GFP (MMRRC:031901) activities were essentially confined to fetal chondrocytes within cartilage templates, with Fgfr3-GFP[+] cells representing a more differentiated subset of SOX9[+] cells located toward the center of the cartilage template (Fig. 1f). Osx-mCherry activities were observed not only in the perichondrium but also in the cartilage template representing an even more differentiated subset of SOX9[+]Fgfr3-GFP[+] chondrocytes. Additionally, while *Col2a1* mRNA was almost exclusively expressed within the cartilage template, *Dlx5* mRNA was predominantly expressed in the perichondrium (Fig. 1g and Supplementary Fig. 1c). Therefore, these in vivo gene expression patterns are concordant with our cluster designation of chondrocytes and perichondrial cells in the scRNA-seq dataset.

To predict the putative cells-of-origins in the fetal cartilage and the perichondrium, we subsequently performed RNA velocity analysis[26]. Two putative cells-of-origins were computationally predicted; (1) the edge of Cluster 1 corresponding to relatively mature chondrocytes, and (2) the midpoint between Clusters 4 and 5 corresponding to relatively immature perichondrial cells (Fig. 1h). Tracing these velocity vectors backward using CellRank inferred Cluster 5 as a putative origin for the differentiation process (Fig. 1i).

Therefore, these computational analyses revealed that perichondrial cells immediately outside of the osteogenic perichondrium might provide a putative cell-of-origin of the fetal chondrocyte-perichondrial cell lineage. We focused on *Dlx5* and *Fgfr3* as markers for early perichondrial cells and fetal chondrocytes, respectively, in subsequent in vivo lineage-tracing studies.

### Dlx5[+] ePCs: Dlx5-creER marks early perichondrial cells

We utilized tamoxifen-inducible *Dlx5-creER*[27] and *Osx-creER*[4] lines to mark fetal perichondrial cells and an *Fgfr3-creER* P1-derived artificial chromosome (PAC) transgenic line[28] to mark fetal chondrocytes. These lines were crossed with *R26R*[tdTomato] and *Col1a1(2.3 kb)-GFP* reporters, and to visualize the corresponding cell types, the triple transgenic mice were pulsed at E12.5 and analyzed after 24 h at E13.5.

We first defined the characteristics of cells marked by *Dlx5-creER*, *Osx-creER* and *Fgfr3-creER* at this stage (Dlx5[CE]-E12.5, Osx[CE]-E12.5 and Fgfr3[CE]-E12.5 cells, respectively). Dlx5[CE]-E12.5 cells were located in the outer perichondrium, which was outside of the SOX9[+] cartilage template and Col1a1(2.3 kb)-GFP[+] osteogenic perichondrium; in fact, Dlx5[CE]-E12.5 cells only minimally overlapped with SOX9[+] or Col1a1(2.3 kb)-GFP[+] cells (Fig. 2a). Importantly, Dlx5[CE]-E12.5 cells did not overlap with MYH3[+] skeletal muscle cells or EMCN[+] endothelial cells, demonstrating their perichondrial identities (Fig. 2b). In contrast, Osx[CE]-E12.5 cells occupied the inner perichondrium, and adjoined or overlapped with Col1a1(2.3 kb)-GFP[+] cells (Fig. 2a, c). Additionally, many Osx[CE]-E12.5 cells were located within the pre-hypertrophic layer of cartilage templates as reported previously[4]. Fgfr3[CE]-E12.5 cells were located within cartilage templates but not in the outer perichondrium (Fig. 2a).

We further quantified Dlx5[CE]-E12.5, Osx[CE]-E12.5 and Fgfr3[CE]-E12.5 tdTomato[+] cells in the perichondrium and the cartilage template. First, we quantified Col1a1(2.3 kb)-GFP[neg]tdTomato[+] perichondrial cells in the outer perichondrium. Dlx5[CE]-E12.5 cells accounted for approximately half of Col1a1-GFP[neg] perichondrial cells (55.6 ± 4.0%), while Osx[CE]-E12.5 or Fgfr3[CE]-E12.5 cells overlapped only minimally with these cells (Osx[CE]-E12.5: 6.8 ± 1.0%, Fgfr3[CE]-E12.5: 2.1 ± 0.7%, Fig. 2c, left panel). Second, we quantified Col1a1(2.3 kb)-GFP[+]tdTomato[+] osteogenic perichondrial cells in the inner perichondrium. A majority of osteogenic perichondrial cells was composed of Osx[CE]-E12.5 cells (73.1 ± 9.5%), while Dlx5[CE]-E12.5 or Fgfr3[CE]-E12.5 cells were minimally present in this

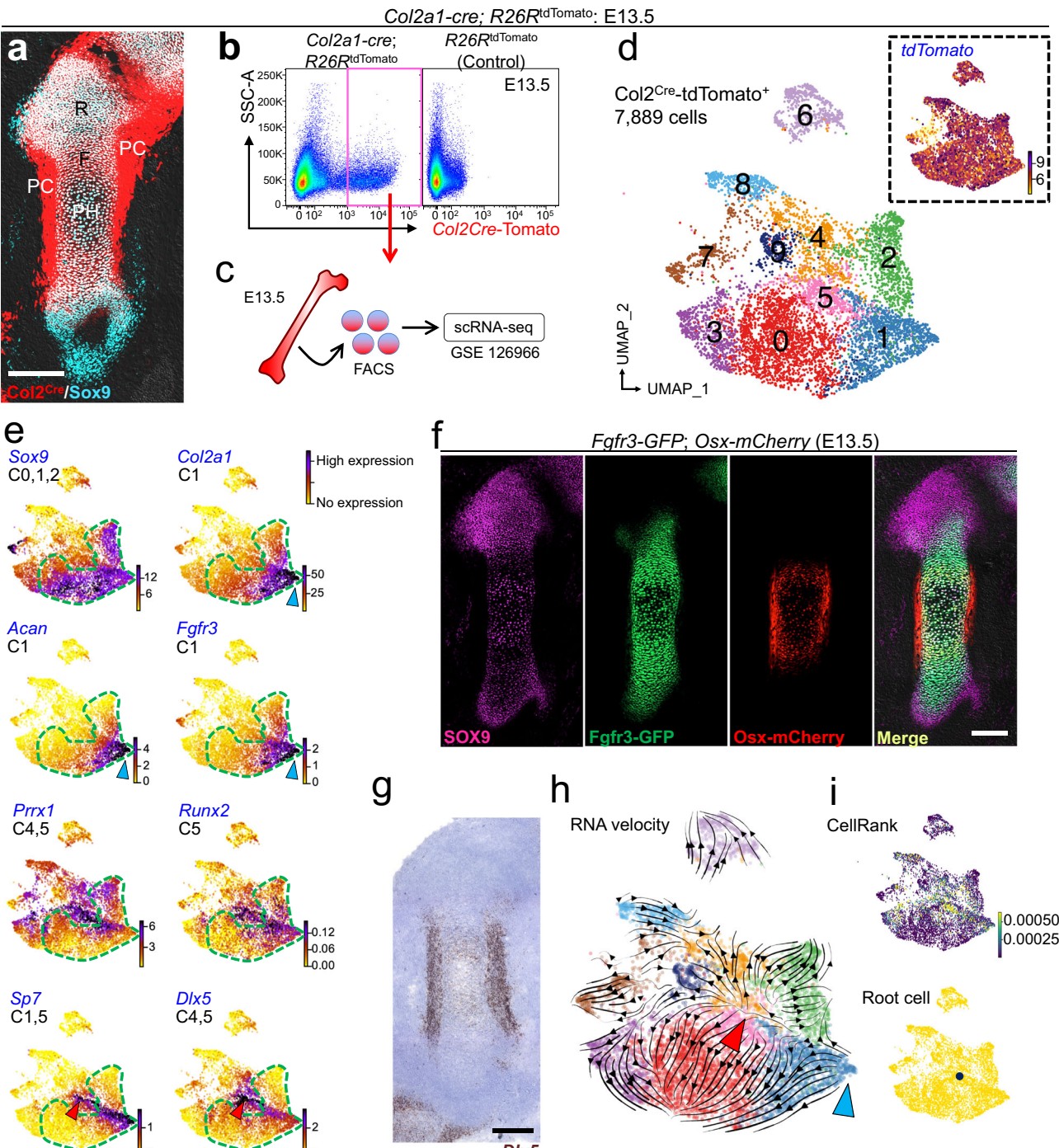

**Fig. 1 | Single-cell RNA-seq identifies a perichondrial cluster as a putative cell origin in the fetal cartilage.** a–e Single-cell RNA-seq analysis of *Col2a1-cre*-marked chondrocytes and perichondrial cells at E13.5. **a**: *Col2a1-cre; R26R*tdTomato femur immunostained for SOX9. R round layer, F flat layer, PH prehypertrophic layer, PC perichondrium, Grey DIC. Scale bar: 200 μm. *n* = 3 mice. **b, c** FACS (**b**) and scRNA-seq (**c**) strategy for Col2a1cre-tdTomato+ cells (red box). Shown are cells isolated from *Col2a1-cre; R26R*tdTomato limbs (left) or *R26R*tdTomato control limbs (right). **d** UMAP plot of major classes of Col2a1cre-tdTomato+ cells (Cluster 0 – 9, 7,889 cells). Pooled from *n* = 5 mice. Dotted box: Feature plot of *tdTomato*. **e** Feature plots of representative genes enriched in each cluster. Cluster 0,1,2: *Sox9*+ (green dotted

contour), Cluster 1: *Col2a1*+, *Acan*+, *Fgfr3*+, Cluster 1,5: *Sp7*+, Cluster 4,5: *Prrx1*+, *Dlx5*+, Cluster 5: *Runx2*+. Violet: high expression, yellow: low expression. **f** *Fgfr3-GFP*; *Osx-mCherry* femur immunostained for SOX9. Scale bar: 200 μm. *n* = 4 mice. **g** RNAScope analyses of *Dlx5*. Scale bar: 200 μm. *n* = 4 mice. **h** RNA velocity analysis. Dynamical model-based RNA velocity vectors superimposed on the UMAP plot. The origins of black arrows represent the inferred initial states, namely Cluster 1 (blue arrowhead) and Cluster 5 (red arrowhead). **i** Initial state and root cell inference. Top: colored by GPCCA-based, CellRank-computed initial state probability. Violet indicates low probability while yellow indicates high probability. Bottom: Cluster 5 is the inferred root cell population.

area (Dlx5CE-E12.5: 9.9 ± 0.7%, Fgfr3CE-E12.5: 6.2 ± 2.6%, Fig. 2c, center panel). Third, we quantified SOX9+tdTomato+ chondrocytes in the cartilage template proper. As expected, Fgfr3CE-E12.5 cells constituted a substantial fraction of SOX9+tdTomato+ chondrocytes, and OsxCE-

E12.5 cells composed a smaller but a sizable fraction of these cells. In contrast, virtually no Dlx5CE-E12.5 cells constituted these chondrocytes (Dlx5CE-E12.5: 2.2 ± 0.4%, OsxCE-E12.5: 18.0 ± 3.9%, Fgfr3CE-E12.5: 36.6 ± 1.8%, Fig. 2c, right panel).

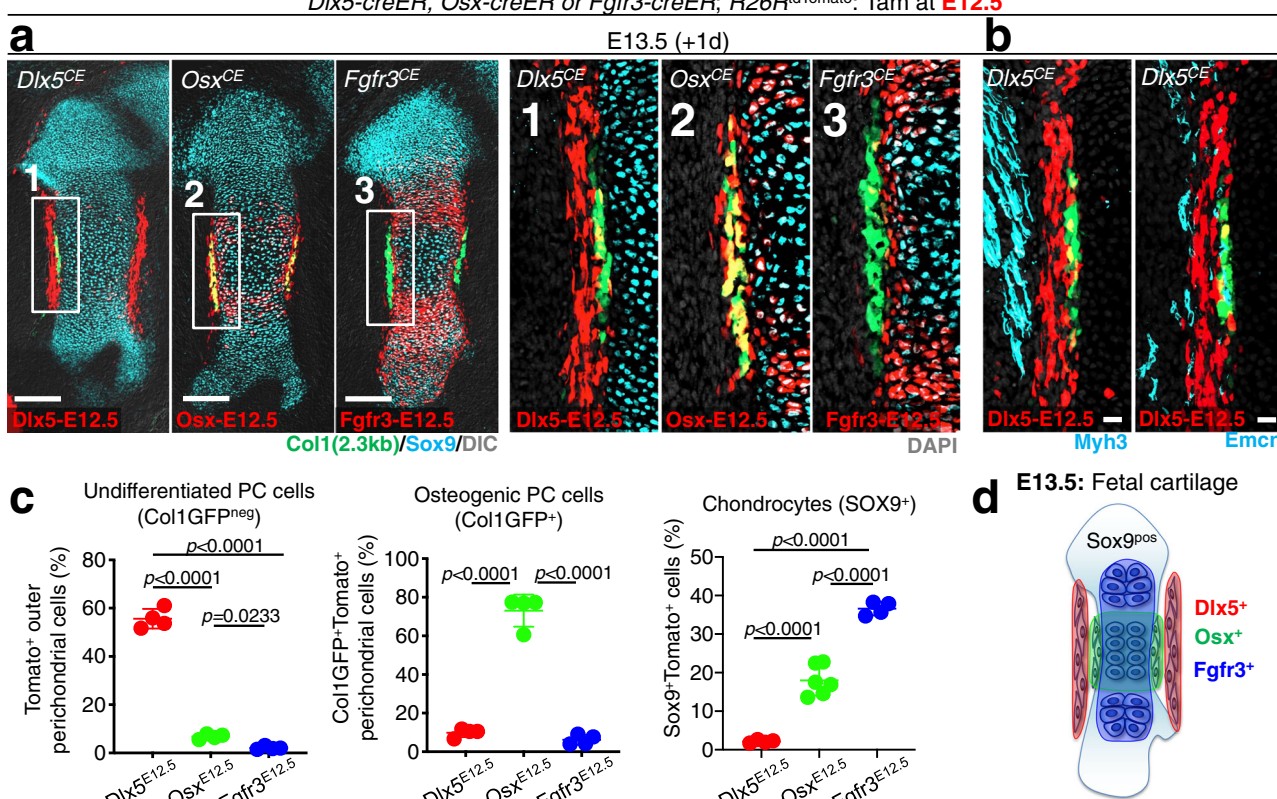

**Fig. 2 | *Dlx5-creER* marks early perichondrial cells of the fetal cartilage.**
**a–d** Localization of Dlx5-creER⁺, Osx-creER⁺, or Fgfr3-creER⁺ cells at E13.5 (pulsed at E12.5), visualized by *cre*-inducible *R26R*^tdTomato reporter, with *Col1a1(2.3 kb)-GFP* reporters to mark osteoblasts. **a** E13.5 cartilage template immunostained for SOX9. Left panels: Scale bar: 200 μm. Right panels: magnified view of the boxed areas (1–3). *n* = 4 mice per each group. **b** *Col1a1(2.3 kb)-GFP; Dlx5-creER; R26R*^tdTomato perichondria at E13.5, immunostained for MYH3 (skeletal muscles, left) or EMCN (endothelial cells, right). Grey: DAPI. Scale bar: 20 μm. *n* = 4 mice per each group. **c** Quantification of Dlx5-creER⁺, Osx-creER⁺, or Fgfr3-creER⁺tdTomato⁺ cells. Left: percentage of tdTomato⁺ cells among Col1a1-GFP^neg perichondrial cells. Center: percentage of Col1a1-GFP⁺tdTomato⁺ cells among Col1a1-GFP⁺ osteogenic perichondrial cells. Right: percentage of SOX9⁺tdTomato⁺ cells among SOX9⁺ chondrocytes. *n* = 4 mice per each group. Two-tailed, one-way ANOVA followed by Tukey's post-hoc test. Data are presented as mean ± s.d. Exact *P* value is indicated in the figures. **d** *Dlx5-creER* and *Fgfr3-creER* can mark mutually exclusive cell populations in the fetal perichondrium and cartilage template, respectively, whereas *Osx-creER* marks cells that overlap with those two cell types. Source data are provided as a Source Data file.

Therefore, *Dlx5-creER* can preferentially mark Col1a1-GFP negative cells in the outer perichondrium, which are distinct from osteogenic perichondrial cells or chondrocytes marked by *Osx-creER* or *Fgfr3-creER*, respectively. In other words, *Dlx5-creER* and *Fgfr3-creER* can mark mutually exclusive cell populations in the fetal perichondrium and cartilage template, respectively, whereas *Osx-creER* marks cells that overlap with those two cell types (Fig. 2d).

**Dlx5⁺ ePCs contribute to cortical bone and marrow stroma**
Subsequently, we traced the fate of Dlx5-creER⁺ early perichondral cells during the formation of the primary ossification center. After 3 days of chase at E15.5, Dlx5^CE-E12.5 cells expanded and contributed to the perichondrium and the newly formed bone collar (periosteum), while only minimally contributing to the marrow space (Fig. 3a-1). In contrast, Osx^CE-E12.5 and Fgfr3^CE-E12.5 cells populated the nascent marrow space in large numbers (Fig. 3a–2,3), indicating that the nascent marrow space may be initially occupied by the cells derived from the cartilage template. The apparently higher contribution of Osx^CE-E12.5 cells to the marrow space than that of Fgfr3^CE-E12.5 cells at this stage likely reflects the fact that *Osx-creER* marks pre-hypertrophic and hypertrophic chondrocytes that can promptly move into the marrow space, while *Fgfr3-creER* marks proliferating chondrocytes that take some time to move into the marrow space. Importantly, Osx^CE-E12.5 cells disappeared from the perichondrium and the pre-hypertrophic layer after translocation (Fig. 3a-2), while Fgfr3^CE-E12.5 cells expanded

within the cartilage template (Fig. 3a-3), highlighting that *Fgfr3-creER* can mark chondroprogenitor cells whereas *Osx-creER* marks pre-hypertrophic chondrocytes with transient nature.

After 6 days of chase at E18.5, when the marrow space expanded further and hematopoiesis was established, Dlx5^CE-E12.5 cells contributed to trabecular bone osteoblasts and marrow stromal cells (Fig. 3b-2). Dlx5^CE-E12.5 cells also expanded in the periosteum and the perichondrium, contributing to the groove of Ranvier (Fig. 3b-1, arrowheads) and the cortical bone. In contrast, Fgfr3^CE-E12.5 cells continued to contribute to the marrow space, the growth plate cartilage and part of the perichondrium (Fig. 3b-5,6). Importantly, Osx^CE-E12.5 cells continued to contribute to the marrow space but not to the perichondrium (Fig. 3b-3,4), underscoring their transient nature in the perichondrium as reported previously[29].

Additionally, we defined the presumed migratory path of Dlx5^CE-E12.5 cells from the perichondrium to the marrow space by analyzing serial time points from E13.5 to E18.5. Consistent with our findings described above, Dlx5^CE-E12.5 cells expanded within the Col1a1-GFP^neg outer domain of the perichondrium for two days (Supplementary Fig. 2a, b), and subsequently started to move laterally to the Col1a1-GFP⁺ osteogenic perichondrium after 3 days of chase at E15.5 (Supplementary Fig. 2c). After 4 days of chase at E16.5, Dlx5^CE-E12.5 cells moved further laterally toward the nascent marrow space and massively expanded therein, and progressively differentiated into Col1a1(2.3 kb)-GFP⁺ osteoblasts of the trabecular and cortical

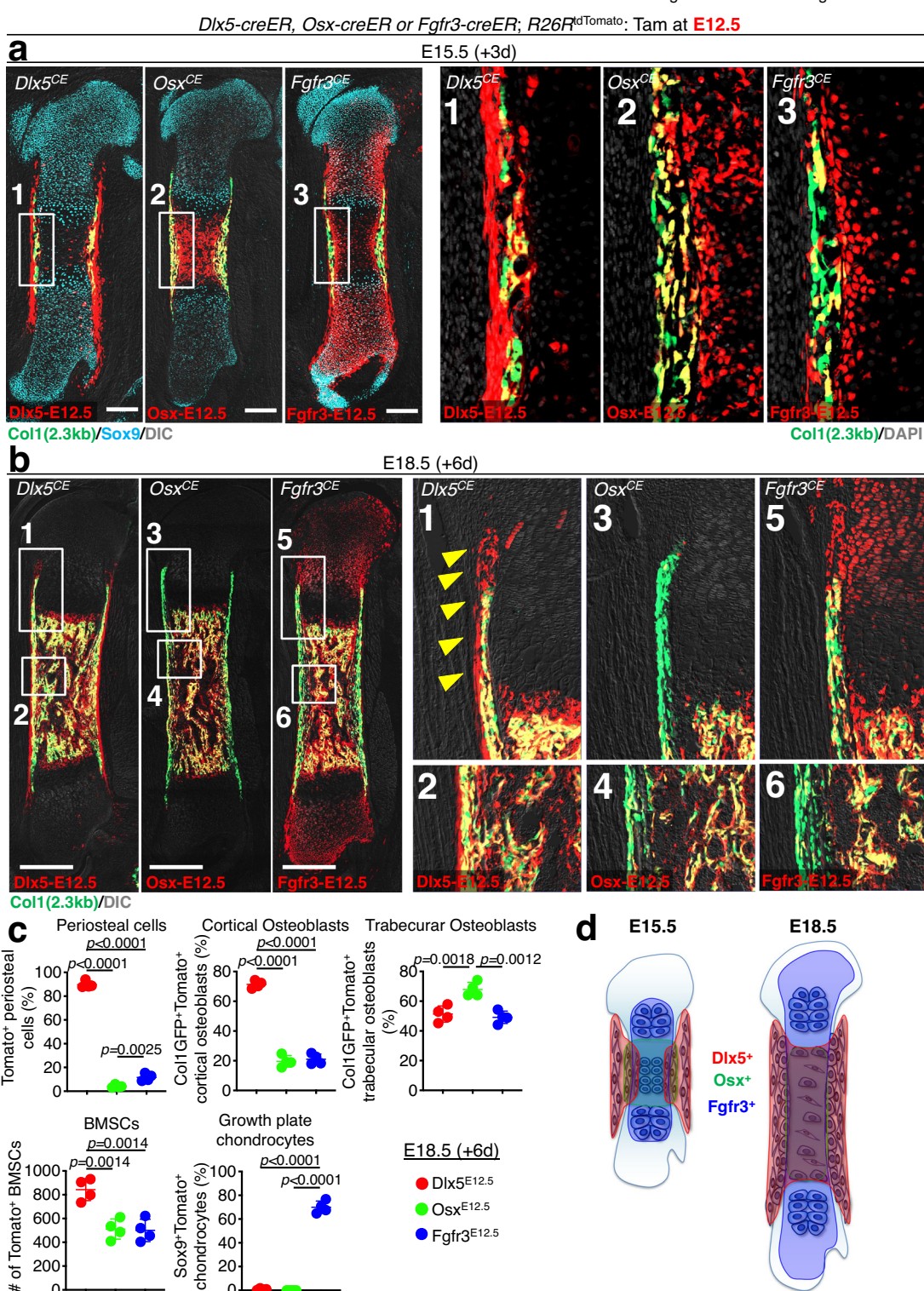

**Fig. caption (embedded in figure):**

E12.5 → E15.5 → E18.5

Dlx5-E12.5: Outer perichondrial cells
Osx-E12.5: Inner perichondrial cells
Fgfr3-E12.5: Chondrogenic cells

*Dlx5-creER, Osx-creER or Fgfr3-creER; R26R^tdTomato: Tam at E12.5*

**a** E15.5 (+3d)

Dlx5^CE | Osx^CE | Fgfr3^CE

Dlx5-E12.5 | Osx-E12.5 | Fgfr3-E12.5
Col1(2.3kb)/Sox9/DIC

Dlx5-E12.5 | Osx-E12.5 | Fgfr3-E12.5
Col1(2.3kb)/DAPI

**b** E18.5 (+6d)

Dlx5^CE | Osx^CE | Fgfr3^CE

Dlx5-E12.5 | Osx-E12.5 | Fgfr3-E12.5
Col1(2.3kb)/DIC

**c**

Periosteal cells — Tomato+ periosteal cells (%) — p<0.0001, p<0.0001, p=0.0025

Cortical Osteoblasts — Col1 GFP+Tomato+ cortical osteoblasts (%) — p<0.0001, p<0.0001

Trabecular Osteoblasts — Col1 GFP+Tomato+ trabecular osteoblasts (%) — p=0.0018, p=0.0012

BMSCs — # of Tomato+ BMSCs — p=0.0014, p=0.0014

Growth plate chondrocytes — Sox9+Tomato+ chondrocytes (%) — p<0.0001, p<0.0001

E18.5 (+6d)
● Dlx5^E12.5
● Osx^E12.5
● Fgfr3^E12.5

**d** E15.5 | E18.5

Dlx5+ | Osx+ | Fgfr3+

compartments (Supplementary Fig. 2d–f). These findings demonstrate that Dlx5+ perichondrial cells can travel laterally to enter the marrow space.

We further quantified Dlx5^CE-E12.5, Osx^CE-E12.5 and Fgfr3^CE-E12.5 tdTomato+ cells at E18.5 in various bone compartments established by this stage. First, we quantified tdTomato+ cells in the periosteum. Dlx5^CE-E12.5 cells occupied essentially the entire periosteum

(89.7 ± 2.9%), while Osx^CE-E12.5 or Fgfr3^CE-E12.5 cells were only minimally present in this area (Osx^CE-E12.5: 4.1 ± 1.3%, Fgfr3^CE-E12.5: 11.7 ± 3.1%, Fig. 3c, upper left). Second, we quantified Col1a1(2.3 kb)-GFP+tdTomato+ osteoblasts in the cortical bone. Dlx5^CE-E12.5 cells contributed to a large majority of cortical bone osteoblasts (periosteal and endosteal osteoblasts, 71.4 ± 2.5%), while Osx^CE-E12.5 or Fgfr3^CE-E12.5 cells contributed to a much smaller fraction of these cells (Osx^CE-

**Fig. 3 | Dlx5-creER⁺ early perichondrial cells contribute to both cortical and marrow stromal compartment. a–c** Cell-fate analysis of Dlx5-creER⁺, Osx-creER⁺, or Fgfr3-creER⁺ cells at E15.5 (pulsed at E12.5), visualized by *cre*-inducible *R26R*-tdTomato reporter, with *Col1a1(2.3 kb)-GFP* reporters for osteoblasts. **a** E15.5 whole femurs. Scale bar: 200 μm. Right panels: magnified view of the boxed areas. *n* = 4 mice per each group. **b** E18.5 whole femurs. Right panels: magnified view of the boxed areas showing perichondrium, growth plate and bone collar. Arrowhead: groove of Ranvier and periosteum. Scale bar: 500 μm. *n* = 4 mice per each group. **c** Quantification of lineage-marked tdTomato⁺ cells. Upper left: percentage of tdTomato⁺ cells among total periosteal cells. Upper center: percentage of Col1a1-

GFP⁺tdTomato⁺ cells among Col1a1-GFP⁺ cortical osteoblasts. Upper right: percentage of Col1a1-GFP⁺tdTomato⁺ cells among Col1a1-GFP⁺ trabecular osteoblasts. Lower left: number of tdTomato⁺ BMSCs. Lower right: percentage of SOX9⁺tdTomato⁺ cells among SOX9⁺ growth plate chondrocytes. *n* = 4 mice per each group. Two-tailed, one-way ANOVA followed by Tukey's post-hoc test. Data are presented as mean ± s.d. Exact *P* value is indicated in the figures. **d** Diagram depicting lineage contribution of Dlx5-creER⁺, Osx-creER⁺, and Fgfr3-creER⁺ cells in fetal endochondral bone development, at E15.5 and E18.5. Source data are provided as a Source Data file.

E12.5: 19.6 ± 3.8%, Fgfr3ᶜᴱ-E12.5: 21.0 ± 3.5%, Fig. 3c, upper center), indicating that a majority of cortical bone osteoblasts are derived from Dlx5⁺ perichondrial cells. Third, we quantified Col1a1(2.3 kb)-GFP⁺ tdTomato⁺ osteoblasts in the trabecular bone. Importantly, Dlx5ᶜᴱ-E12.5 perichondrial cells and Fgfr3ᶜᴱ-E12.5 chondrocytes contributed to an equivalent fraction of trabecular bone osteoblasts (Dlx5ᶜᴱ-E12.5: 51.3 ± 5.4%, Fgfr3ᶜᴱ-E12.5: 49.2 ± 4.2%, Fig. 3c, upper right), indicating that trabecular bone osteoblasts at this stage might be equally derived from the two cells-of-origins of Dlx5⁺ perichondrial cells and Fgfr3⁺ chondrocytes. We also found that Osxᶜᴱ-E12.5 cells contributed to a large fraction of these cells at this stage (67.9 ± 4.8%), indicating that Osxᶜᴱ-E12.5 cells overlap extensively both with Dlx5ᶜᴱ-E12.5 and Fgfr3ᶜᴱ-E12.5 cells. Fourth, we quantified tdTomato⁺ bone marrow stromal cells (BMSCs) within the marrow space. Dlx5ᶜᴱ-E12.5 cells contributed to a higher number of BMSCs in the marrow space than the other cell types do (Dlx5ᶜᴱ-E12.5: 842.7 ± 91.9 cells, Osxᶜᴱ-E12.5: 511.0 ± 84.8 cells, Fgfr3ᶜᴱ-E12.5: 500.0 ± 94.0 cells, Fig. 3c, lower left), indicating that Dlx5⁺ early perichondrial cells may preferentially contribute to marrow stromal cells. Fifth, we quantified SOX9⁺tdTomato⁺ chondrocytes with the growth plate. As expected, only Fgfr3ᶜᴱ-E12.5 cells, but not Dlx5ᶜᴱ-E12.5 or Osxᶜᴱ-E12.5 cells, contributed to growth plate chondrocytes at this stage (Fgfr3ᶜᴱ-E12.5: 70.0 ± 5.1%, Dlx5ᶜᴱ-E12.5: 1.0 ± 0.7%, Osxᶜᴱ-E12.5: 0%, Fig. 3c, lower right), demonstrating that Dlx5⁺ early perichondrial cells do not become growth plate chondrocytes.

Therefore, trabecular bone osteoblasts and BMSCs within the marrow space have dual origins in the fetal perichondrium and the cartilage template among Dlx5-creER⁺ cells and Fgfr3-creER⁺ cells, respectively, while Osx-creER⁺ cells overlap with these cell types. In contrast, periosteal cells are almost entirely formed from the outer perichondrium, and growth plate chondrocytes are entirely formed from the cartilage template (Fig. 3d).

## Dlx5⁺ ePCs regulate marrow formation via Hedgehog pathway

We next investigated molecular mechanisms that coordinate the differentiation of Dlx5-creER⁺ perichondrial cells and Fgfr3-creER⁺ fetal chondrocytes to achieve the formation of the bone marrow space. To this end, we performed additional computational analyses by CellChat[30] on our scRNA-seq dataset (Figs. 1 and 4 red dotted area) to unravel intercellular interactions. This analysis can identify the signaling pathways regulating the two cells-of-origins present in the fetal cartilage, namely, *Fgfr3*⁺ Cluster 1 and *Dlx5*⁺ Cluster 5.

The overall level of signaling among clusters varied by cell type, with a variety of strengths of outgoing and incoming interactions (Fig. 4a, left panel). Hedgehog (Hh) Pathways showed strong outgoing and incoming interaction strengths in *Fgfr3*⁺ Cluster 1, and incoming interaction strength in *Dlx5*⁺ Cluster 5 (Fig. 4a, right panel), indicating potential functional significance of Hh pathways in regulating the two cells-of-origins. This could be explained by the fact that *Ihh* was most strongly expressed in Cluster 1, whereas its cognate receptor *Ptch1* was expressed both in Cluster 1 and 5 (Fig. 4b, c). The CellChat analysis further revealed that Cluster 1 was classified as the most important sender, receiver and influencer of Hh signaling, while Cluster 5 was listed as the second most important receiver (Fig. 4d, e). Among genes in the Hh signaling pathway, *Indian hedgehog* (*Ihh*) was identified as an

autocrine signaling factor, expressed by and targeting *Ptch1*⁺ cells in Cluster 1 (Fig. 4f). The *Ihh-Ptch1* paracrine signaling originating from Cluster 1 targeted the other clusters (Fig. 4e, f); among these interactions, the paracrine signaling association between Cluster 1 and Cluster 5 was the strongest, wherein Cluster 5 received the strongest signals from Cluster 1 (Fig. 4f). Therefore, *Ihh-Ptch1* signaling was predicted as an important pathway mediating intercellular interactions between the fetal perichondrium and cartilage template via paracrine regulations.

We further performed functional analyses of the Hedgehog signaling pathway regulating the differentiation of the two cells-of-origins types i.e. Dlx5⁺ early perichondrial cells and Fgfr3⁺ fetal chondrocytes. To this end, we conditionally deleted *Ptch1* in these cells using *Ptch1*-floxed alleles. *Dlx5-creER; Ptch1*ᶠˡ/⁺*; R26R*ᵗᵈᵀᵒᵐᵃᵗᵒ (Dlx5-Control) and *Dlx5-creER; Ptch1*ᶠˡ/ᶠˡ*; R26R*ᵗᵈᵀᵒᵐᵃᵗᵒ (Dlx5-Ptch1 cKO) mice, as well as *Fgfr3-creER; Ptch1*ᶠˡ/⁺*; R26R*ᵗᵈᵀᵒᵐᵃᵗᵒ (Fgfr3-Control) and *Fgfr3-creER; Ptch1*ᶠˡ/ᶠˡ*; R26R*ᵗᵈᵀᵒᵐᵃᵗᵒ (Fgfr3-Ptch1 cKO) mice were pulsed at E12.5, and analyzed at E18.5. Deletion of *Ptch1* leads to constitutive activation of Hedgehog signaling in these cells[31–33].

Interestingly, the whole femur length was not altered in the both mutant (Dlx5-Ptch1 cKO and Fgfr3-Ptch1 cKO) mice. However, the bone marrow length relative to the total length was significantly reduced in both Dlx5-Ptch1 cKO and Fgfr3-Ptch1 cKO mice compared to Control (Fig. 4g). This reduction was more pronounced in Fgfr3-Ptch1 cKO mice, associated with an increase of the distal cartilage length relative to the total length, suggesting a delay in the cartilage-to-marrow stroma transition. In contrast in Dlx5-Ptch1 cKO mice, only the bone marrow length relative to the total length was reduced without increasing the cartilage length (Fig. 4g, second column from the right), supporting the theory that Dlx5⁺ perichondrial cells regulate the marrow formation without involving chondrocytes within cartilage templates. Further, the Col1a1(2.3 kb)-GFP⁺ trabecular area per the marrow area was significantly reduced in Fgfr3-Ptch1 cKO, associated with an increase in the number of chondrocytes in each column while the total number of chondrocyte columns was unchanged (Fig. 4g, first column from the right), further suggesting a delay in the chondrocyte-to-osteoblast transition.

Therefore, Dlx5⁺ early perichondrial cells directly regulate the formation of the marrow space, whereas Fgfr3⁺ fetal chondrocytes regulate bone formation within the marrow space through chondrocyte-to-osteoblast transition, at least in part mediated through Hedgehog signaling.

## Dlx5⁺ ePCs contribute to postnatal diaphyseal marrow stroma

We further set out to define how Dlx5-creER⁺ early perichondrial cells contribute postnatally to the bone and marrow stromal compartment. For this purpose, we analyzed the cell fates of Dlx5ᶜᴱ-E12.5, Osxᶜᴱ-E12.5 and Fgfr3ᶜᴱ-E12.5 cells after 4 weeks of chase at postnatal day (P) 21.

As expected, Dlx5ᶜᴱ-E12.5 cells substantially contributed to the postnatal cortical bone compartment. These cells became undifferentiated periosteal cells and Col1a1(2.3 kb)-GFP⁺ cortical bone osteoblasts (Fig. 5a, b and Supplementary Fig. 3a). However, in bone marrow, Dlx5ᶜᴱ-E12.5 cells contributed only to the diaphyseal marrow stromal compartment. These cells became Col1a1(2.3 kb)-GFP⁺ trabecular bone osteoblasts and Cxcl12-GFPʰⁱᵍʰ reticular stromal cells in the

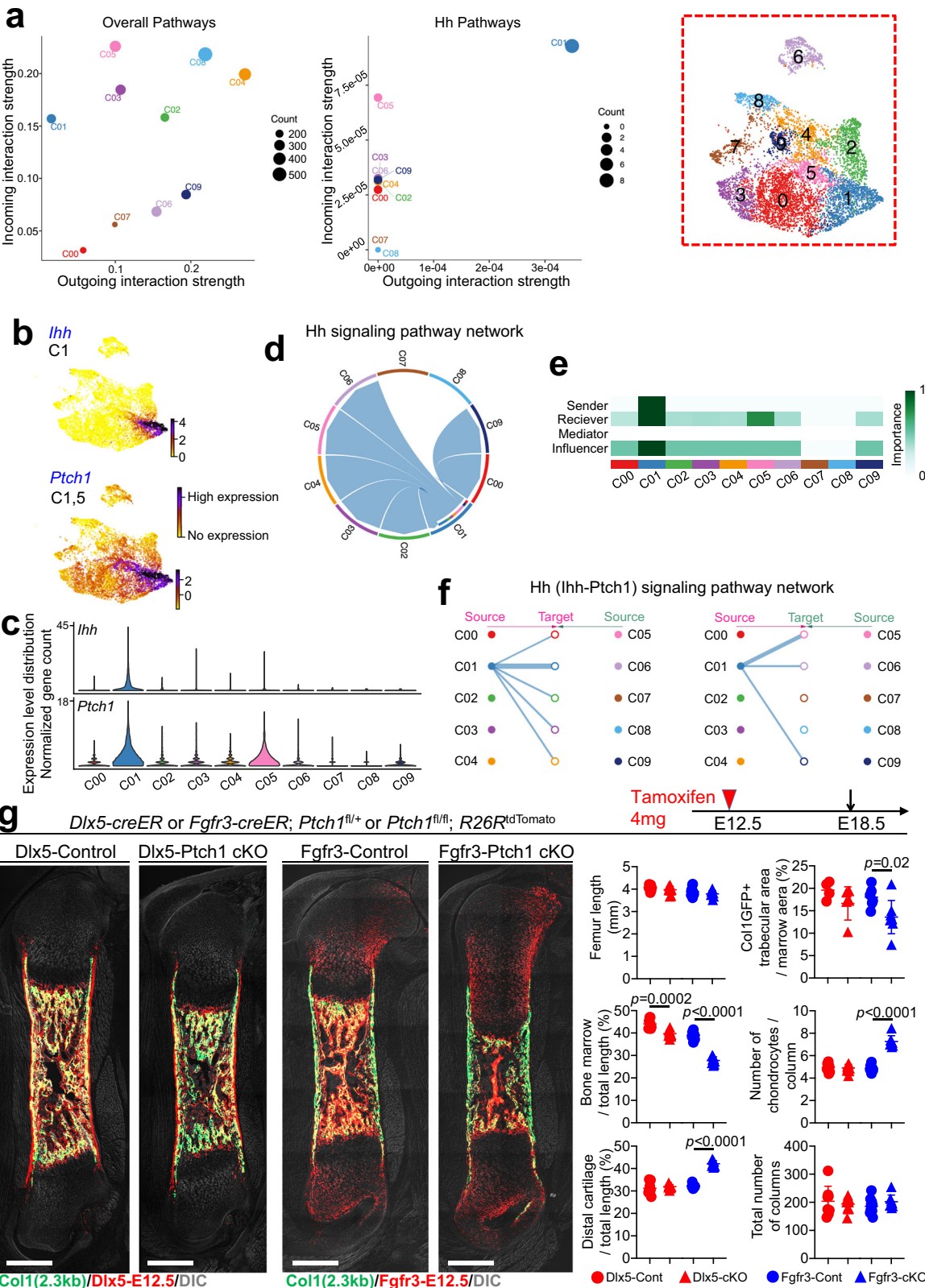

diaphysis, but did not generate the same cell types in the metaphysis (Fig. 5a, b and Supplementary Fig. 3a). In contrast, Osx[CE]-E12.5 cell-derived osteoblasts and marrow stromal cells largely disappeared by this stage due to their transient nature[6,22]. Fgfr3[CE]-E12.5 cells became cortical and trabecular osteoblasts as well as Cxcl12-GFP[high] stromal cells predominantly in the metaphyseal marrow space, seemingly in a mutually exclusive manner to Dlx5[CE]-E12.5 cell derivatives (Fig. 5a, b

and Supplementary Fig. 3a). Interestingly, both Dlx5[CE]-E12.5 cells and Fgfr3[CE]-E12.5 cells had disappeared from the perichondrium (groove of Ranvier) by this stage, and appeared to make only minimal contribution to the endosteum.

We further defined how Dlx5[CE]-E12.5 cells contributed to the developing bone marrow space and cortical bone using quantitative histological approaches. Dlx5[CE]-E12.5 perichondrial cells predominantly

**Fig. 4 | Dlx5-creER⁺ early perichondrial cells regulate marrow formation in response to Hedgehog signaling. a–f** CellChat intercellular communication analysis of the E13.5 *Col2a1-cre*-marked chondrocyte and perichondrial cell scRNA-seq dataset. **a** Outgoing and incoming interaction strength of signaling pathways. Each circle represents the number of interactions a cluster participates in. (Left): Cluster 1 has a low tendency to send signals but a high tendency to receive signals in overall pathways. (Right): Cluster 1 exhibits a high propensity to both send and receive signals via the Hh pathways. Dotted box: UMAP-based plot and cluster labels of the E13.5 scRNA-seq data. **b** Feature plots of *Ihh* and *Ptch1*. Violet: high expression, yellow: low expression. **c** Violin plots displaying the expression distribution of *Ihh* and *Ptch1* in each cluster. **d** Chord diagram shows the signals linking Cluster 1 and other clusters via the Hh pathway. **e** Heatmap represents the relative importance of each cluster. Dark green: higher importance, white: lower importance. Cluster 1 is scored as the dominant sender, receiver, and influencer of the Ihh-Ptch1 signaling pathway. Cluster 5 is measured as a dominant receiver. **f** Hierarchical plot showing the intercellular interactions of the chondrocyte-perichondrial cell lineage via the Ihh-Ptch1 signaling pathway. **g** Functional analysis of Hedgehog signaling in Dlx5-creER⁺ early perichondrial cells and Fgfr3-creER⁺ fetal chondrocytes in bone marrow development. Left panels: Histological images of *Dlx5-creER; Ptch1*fl/+; *R26R*tdTomato (Dlx5-Control), *Dlx5-creER; Ptch1*fl/fl; *R26R*tdTomato (Dlx5-Ptch1 cKO), *Fgfr3-creER; Ptch1*fl/+; *R26R*tdTomato (Fgfr3-Control), *Fgfr3-creER; Ptch1*fl/fl; *R26R*tdTomato (Fgfr3-Ptch1 cKO), femur at E18.5 (pulsed at E12.5). Scale bar: 500 μm. *n* = 8 (Dlx5-Control), *n* = 11 (Dlx5-Ptch1 cKO), *n* = 10 (Fgfr3-Control), *n* = 8 (Dlx5-Ptch1 cKO) mice per each group. Right panels: quantification of whole femur length (top left), bone marrow length per total bone length (middle left), distal cartilage length per total bone length (bottom left), Col1a1-GFP⁺ trabecular area per total bone marrow area (top right), number of chondrocytes per column (middle right) and total number of columns in growth plate (bottom right). *n* = 8 (Dlx5-Control), *n* = 11 (Dlx5-Ptch1 cKO), *n* = 10 (Fgfr3-Control), *n* = 8 (Dlx5-Ptch1 cKO) mice per group. Two-tailed, Mann–Whitney's *U* test. Data are presented as mean ± s.d. Exact *P* value is indicated in the figures. Source data are provided as a Source Data file.

contributed to BMSCs and osteoblasts and osteocytes in the distal area (3–7 mm from the growth plate), whereas Fgfr3CE-E12.5 cells contributed to these cells in the proximal area (0–3 mm from the growth plate, Fig. 5c). OsxCE-E12.5 cells showed a similar pattern to Dlx5CE-E12.5 cells, but contributed to BMSCs and osteoblasts to a much lesser extent.

Additionally, we performed flow cytometry analysis of cells isolated from femurs of triple transgenic mice carrying a *Cxcl12-GFP* reporter (Fig. 5d and Supplementary Fig. 4) for further quantification. Importantly, Dlx5CE-E12.5 contributed to a substantial fraction of Cxcl12-GFPhigh reticular stromal cells, at a level fewer than Fgfr3CE-E12.5 cells (Dlx5CE-E12.5: 19.5 ± 3.1%, Fgfr3CE-E12.5: 26.4 ± 3.7%). The remaining fraction of Cxcl12-GFPhigh stromal cells might be derived from other perichondrial cells or chondrocytes at E12.5 that were not marked either by *Dlx5-creER* or *Fgfr3-creER*. In contrast, OsxCE-E12.5 cells made essentially no contribution to Cxcl12-GFPhigh reticular stromal cells at this stage (OsxCE-E12.5: 1.0 ± 0.4%). Similarly, not all the osteocytes were derived from either Dlx5CE-E12.5 or Fgfr3CE-E12.5 cells, suggesting that other perichondrial cells or chondrocytes may contribute to the remaining fraction.

The differential contribution of Dlx5-creER⁺ perichondrial cells and Fgfr3-creER⁺ chondrocytes continued further onto the adult stage. At three months of age, Dlx5CE-E12.5 cells were present in the diaphyseal marrow stromal compartment and contributed to Perilipin⁺ marrow adipocytes, while Fgfr3CE-E12.5 cells were present in the metaphyseal marrow stromal compartment and contributed to alkaline phosphatase (ALP)⁺ osteoblast-like cells on the bone surface (Supplementary Fig. 3b). Only a small number of OsxCE-E12.5 cells persisted in the diaphyseal marrow space at this stage (Fig. 5e).

These findings demonstrate that Dlx5-creER⁺ early perichondrial cells make sustained contribution to diaphyseal bone and marrow stromal compartment in a manner distinct from Fgfr3-creER⁺ fetal chondrocytes (Supplementary Fig. 3c).

We also defined how the two cell types (Dlx5-creER⁺ perichondrial cells and Fgfr3-creER⁺ chondrocytes) are related to previously identified chondrogenic cell types marked by *Col2a1-creER*[22,34,35]. Analysis of *Col2a1-creER; R26R*tdTomato mice at E13.5 after 24 h of pulse at E12.5 revealed that Col2a1CE-E12.5 cells were present not only in the cartilage template but also in the perichondrium (Supplementary Fig. 5a). After 2 weeks of chase at P7, Col2a1CE-E12.5 cells contributed to essentially all perichondrial cells and growth plate chondrocytes, as well as to cells of the metaphyseal periosteum and cortical and marrow stromal components; this contribution was far broader than that of Fgfr3CE-E12.5 or Dlx5CE-E12.5 cells (Supplementary Fig. 5b). The same trend continued after 1 month of chase at P21 (Supplementary Fig. 5c). Therefore, *Col2a1-creER* marks both early perichondrial cells and chondrocytes at the fetal cartilage, demonstrating the utility of *Dlx5-creER* and *Fgfr3-creER* to deconvolute the contribution of the chondrocyte-perichondrial cell lineage.

## Adipocyte-biased state of Dlx5⁺ ePC-derived BMSCs

Our findings that Dlx5-creER⁺ early perichondrial cells and Fgfr3-creER⁺ chondrocytes contributed to the different bone marrow stromal compartments (diaphyseal and metaphyseal, respectively) prompted us to investigate molecular differences of the two types of (perichondrium-derived and cartilage-derived) BMSCs. To reveal the unique molecular signature of Dlx5-creER⁺ cell-derived BMSCs, we performed a comparative bulk RNA-seq analysis of Dlx5CE-E12.5 and Fgfr3CE-E12.5 Cxcl12-GFPhigh cells at P21. Cxcl12-GFPhightdTomato⁺ cells were isolated by FACS from *Cxcl12*GFP/+; *Dlx5-creER; R26R*tdTomato and *Cxcl12*GFP/+; *Fgfr3-creER; R26R*tdTomato bones at P21 (pulsed at E12.5; Fig. 6a).

Principal component analysis (PCA) revealed that biological replicates of Dlx5CE-E12.5 and Fgfr3CE-E12.5 Cxcl12-GFPhigh cells grouped by lineage (Fig. 6b, Dlx5CE-E12.5: circles, 1–3, Fgfr3CE-E12.5: triangles, 1–2), demonstrating the unique molecular characteristics of these two cell types. Accordingly, a large number of genes were differentially expressed between the two cell types, with ~1400 and ~1800 genes upregulated in Dlx5CE-E12.5 and Fgfr3CE-E12.5 cells, respectively (Fig. 6c). Particularly, genes associated with chondrocytes and osteoblasts were upregulated in Fgfr3CE-E12.5 Cxcl12-GFPhigh cells, including *Acan, Col2a1, Fgfr3, Sp7, Col1a1*, and *Dmp1* (Fig. 6d), indicating that these fetal cartilage-derived BMSCs retain a chondrocyte-osteoblast-biased state. In contrast, adipocyte-related genes markers, including *Adipoq, Cxcl12, Kitl, Lepr, Lpl*, and *Ebf3* were upregulated in Dlx5CE-E12.5 Cxcl12-GFPhigh cells, indicating that fetal perichondrium-derived BMSCs possess an adipocyte-biased state (Fig. 6d).

Gene Ontology (GO) enrichment analysis of differentially expressed genes revealed that significant enrichment of biologically relevant GO terms in Dlx5CE-E12.5 cells, including negative regulation of cell migration (GO:0030336) and fat cell differentiation (GO:0045444) and ossification (GO:0001503) (Fig. 6e), revealing the quiescent adipocyte-biased identity of Dlx5⁺ perichondrium-derived BMSCs. Conversely, many cell division-related GO terms [mitotic cell cycle phase transition (GO:0044772), mitotic nuclear division (GO:0140014), and ossification (GO:0001503)] were enriched in Fgfr3CE-E12.5 cells (Fig. 6e), revealing the actively cycling nature of Fgfr3⁺ fetal cartilage-derived BMSCs.

Subsequently, we performed colony forming unit-fibroblast (CFU-F) assays of Dlx5CE-E12.5 and Fgfr3CE-E12.5 cells, by plating bone marrow cells isolated from *Dlx5-creER; R26R*tdTomato or *Fgfr3-creER; R26R*tdTomato bone marrow at P21 (pulsed at E12.5) at a clonal density. Dlx5CE-E12.5 cells contributed to only a small fraction of CFU-Fs, whereas Fgfr3CE-E12.5 cells contributed to a large majority of CFU-Fs (Dlx5CE-E12.5: 13.2 ± 3.1%, Fgfr3CE-E12.5: 57.1 ± 11.3%, Fig. 6f). Flow cytometry analysis revealed that Cxcl2-GFPhigh cells represented 66.2 ± 12.6% and 17.3 ± 4.1% of Dlx5-E12.5 and Fgfr3CE-E12.5 cells, respectively, at P21 (Fig. 5d), which does not appear to correlate with their clonogenic

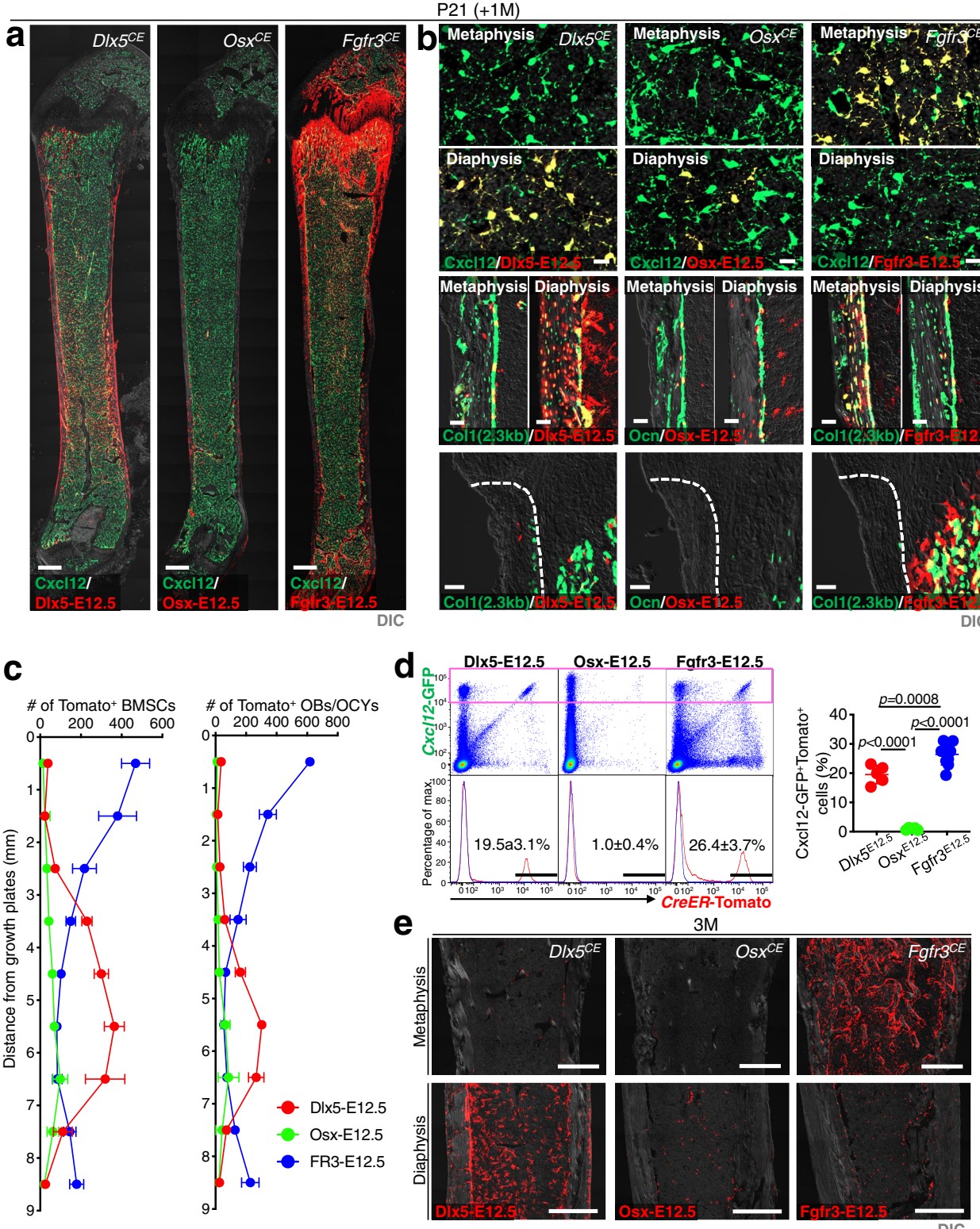

activities. These results are consistent with our recent finding that the adipogenic subset of CXCL12[high] stromal cells (Cxcl12-creER+ stromal cells[36], akin to Adipo-CAR cells[37]) have little colony-forming activities.

Therefore, Dlx5-creER+ perichondrial cell-derived marrow stromal cells possess an adipocyte-biased state, supporting the theory that Dlx5+ early perichondrial cells may preferentially contribute to a pre-adipocyte-like subset of CXCL12+ BMSCs (Adipo-CAR cells) in developing bone marrow (Fig. 6g).

### Anabolic and injury responses of Dlx5+ ePC-derived BMSCs

Lastly, we set out to determine the functional significance of Dlx5-creER+ cell-derived marrow stromal cells in the context of elevated osteogenesis. Because *Pth1r* (encoding PTH/PTHrP receptor) was upregulated in Fgfr3[CE]-E12.5 CXCL12+ BMSCs, we first tested the response of these cells to bone anabolic agents i.e. intermittent PTH administration (iPTH) by giving once daily injections of PTH (1–34) (100 μg/kg b.w.). *Col1a1(2.3 kb)-GFP*; *Dlx5-creER* or *Fgfr3-creER*;

**Fig. 5 | Dlx5-creER⁺ perichondrial cells contribute postnatally to diaphyseal bone marrow stroma. a–d** Contribution of fetal (E12.5) Dlx5-creER⁺, Osx-creER⁺ or Fgfr3-creER⁺ cells to postnatal (P21) skeletal compartments. *Cxcl12^{GFP/+}; Dlx5-creER; R26R^{tdTomato}* ((a): left, (b): upper left), *Cxcl12^{GFP/+}; Osx-creER; R26R^{tdTomato}* ((a): center, (b): upper center), *Cxcl12^{GFP/+}; Fgfr3-creER; R26R^{tdTomato}* ((a): right, (b): upper right), *Col1a1(2.3 kb)-GFP; Dlx5-creER; R26R^{tdTomato}* ((b): middle left, lower left), *Osteocalcin (Ocn) -GFP; Osx-creER; R26R^{tdTomato}* ((b): middle center, lower center) or *Col1a1(2.3 kb)-GFP; Fgfr3-creER; R26R^{tdTomato}* femurs ((b):middle right, lower right). *n* = 4 mice per group. (a): Whole femurs with growth plates on top. Scale bar: 500 μm. *n* = 4 mice per group. (b): (Upper panels): metaphyseal bone marrow (top), diaphyseal bone marrow (bottom). Scale bar: 20 μm. (Middle panels): metaphyseal cortical bone (left), diaphyseal cortical bone (right). Scale bar: 20 μm. Lower panels: groove of Ranvier. Scale bar: 50 μm. (c): Quantification of tdTomato⁺ bone marrow stromal cells (left) and osteoblasts/cytes (right), based on distance from growth plate. Red: Dlx5-creER⁺, Green: Osx-creER⁺, Blue: Fgfr3-creER⁺ cell-derivatives. *n* = 4 mice per each group. (d): Flow cytometry analysis of CD45/Ter119/ CD31^{neg} cells at P21. Bone marrow cells isolated from *Cxcl12^{GFP/+}; Dlx5-creER; R26R^{tdTomato}* femurs (pulsed at E12.5), *Cxcl12^{GFP/+}; Osx-creER; R26R^{tdTomato}* femurs (pulsed at E12.5) and *Cxcl12^{GFP/+}; Fgfr3-creER; R26R^{tdTomato}* femurs (pulsed at E12.5). *n* = 5 (Dlx5-E12.5), *n* = 5 (Osx-E12.5), *n* = 10 (Fgfr3-E12.5) mice per group. Two-tailed, One-way ANOVA followed by Tukey's post-hoc test. Data are presented as mean ± s.d. Exact *P* value is indicated in the figures. **e** Contribution of fetal (E12.5) Dlx5-creER⁺, Osx-creER⁺ or Fgfr3-creER⁺ cells to adult (3 M) skeletal compartments. Upper panels: metaphysis. Lower panels: diaphysis. Scale bar: 500 μm. *n* = 4 mice per each group. Source data are provided as a Source Data file.

*R26R^{tdTomato}* mice (pulsed at E12.5) were treated by iPTH for 3 weeks from P21 to P42 (Fig. 7a).

As expected, this iPTH anabolic regimen substantially increased the trabecular bone mass particularly in the metaphysis associated with a substantial increase in osteoblasts (Fig. 7b, c, left panel) (Col1-GFP⁺ cells: 324.8 ± 40.7 and 362.0 ± 65.0 cells/area for Dlx5^{CE}-E12.5 and Fgfr3^{CE}-E12.5 femurs, respectively, compared to 114.8 ± 11.6 cells/area in Vehicle-treated mice). Importantly, Fgfr3^{CE}-E12.5 cells contributed to a great majority of newly formed osteoblasts, while virtually no Dlx5^{CE}-E12.5 cells contributed to these cells in the metaphysis (Fig. 7c, right panel; GFP⁺tdTomato⁺ cells; 2.4 ± 0.9% and 55.1 ± 5.0% of total Col1-GFP⁺ cells for Dlx5^{CE}-E12.5 and Fgfr3^{CE}-E12.5 femurs, respectively), demonstrating that Dlx5-creER⁺ cell-derived BMSCs barely account for iPTH-induced anabolic responses.

Second, we investigated the capability of Dlx5^{CE}-E12.5 and Fgfr3^{CE}-E12.5 cells for injury-responsive osteogenesis using a femoral bone marrow ablation model, which induced direct differentiation of BMSCs within the marrow cavity (Fig. 7d)[38]. After 7 days of marrow ablation, Dlx5^{CE}-E12.5 cells contributed to de novo bone formation exclusively in the diaphysis, while Fgfr3^{CE}-E12.5 cells contributed predominantly in the metaphysis (Fig. 7e, f; GFP⁺tdTomato⁺ cells; 2.2 ± 1.5% and 76.0 ± 7.9% of total Col1-GFP⁺ cells in metaphyseal bone marrow, 57.4 ± 6.9% and 9.0 ± 3.0% of total Col1-GFP⁺ cells in diaphyseal bone marrow for Dlx5^{CE}-E12.5 and Fgfr3^{CE}-E12.5 femurs, respectively).

Therefore, Dlx5-creER⁺ perichondrial cell-derived BMSCs are refractory to bone anabolism by iPTH, and can contribute locally to reactive osteogenesis in the diaphyseal marrow space. In contrast, Fgfr3-creER⁺ chondrocyte-derived BMSCs have both active anabolic and regenerative osteogenic responses in the metaphyseal marrow space.

## Discussion

Here, we discovered a population of early perichondrial cells in the outer layer of the perichondrium defined by Dlx5 expression. We demonstrate that the Dlx5⁺ early perichondrial cells provide an important cell-of-origin for the diaphyseal bone marrow stroma of postnatal bones (Fig. 8). These Dlx5⁺ early perichondrial cells represent the second wave of cells populating the developing marrow space, which enter the marrow space following the initial wave of fetal chondrocytes that transform into marrow stroma-constituting cells. These perichondrium-derived cells subsequently occupy the diaphyseal bone marrow as stromal cells with an adipocyte-biased state, some of which also become trabecular bone osteoblasts. Our findings are based on the two cell type-specific inducible genetic tools that we identified in this study – *Dlx5-creER* for early perichondrial cells and *Fgfr3-creER* for fetal chondrocytes – which enable the deconvolution of the fetal chondrocyte-perichondrial cell lineage in bone marrow formation.

Interestingly, Dlx5⁺ perichondrial cell-derived marrow stromal cells are characterized by pre-adipocyte-like characteristics and do not respond to bone anabolism by intermittent administration of parathyroid hormone (PTH). The emerging concept is that a subset of bone marrow stromal cells termed CXCL12-abundant reticular stromal cells (CAR cells) are composed of two distinct cellular subsets of pre-osteoblast-like (Osteo-CAR) cells and pre-adipocyte-like (Adipo-CAR) cells[37]. Our recent study demonstrates that Osteo-CAR and Adipo-CAR cells are likely to represent independent cell lineages that do not contribute one another[39]. Our findings provide preliminary evidence that quiescent pre-adipocyte-like marrow stromal (Adipo-CAR) cells in the diaphyseal marrow space might preferentially originate from the fetal perichondrium, while more active pre-osteoblast-like marrow stromal (Osteo-CAR) cells in the metaphyseal marrow space might predominantly originate from the fetal cartilage during bone marrow development.

One of the central questions regarding the two CAR cell types is whether these cells represent the recent bifurcation of a single progenitor present in adults that gives rise to both cell types or whether these are two wholly distinct cell lineages with distinct origins that happen to share some markers in common. The data presented here argues for the latter possibility, although further studies are needed to delineate such differential lineage contribution. Our findings support the theory that the adult bone marrow stromal compartments are developmentally prescribed within the two cells-of-origins of the fetal cartilage.

Recent investigation demonstrates that leptin receptor (LepR)-expressing marrow stromal cells are composed of two distinct peri-arteriolar and peri-sinusoidal subsets with different functionality. The peri-arteriolar LepR⁺ cells express osteolectin (encoded by *Clec11a*), which are rapidly-dividing and short-lived and poised to osteogenesis in a mechanosensitive manner[40]. This osteolection⁺ subset of LepR⁺ cells may coincide with Osteo-CAR cells that are localized to peri-arteriolar surfaces[37]. Importantly, these LepR⁺osteolectin⁺ cells are not fated to adipocytes[40] and Osteo-CAR cells are not computationally predicted to give rise to Adipo-CAR cells[37]. Therefore, these studies also indicate that bone marrow stromal cells poised for osteogenesis and adipogenesis represent two distinct cell lineages. It remains to be determined whether Fgfr3⁺ fetal chondrocyte-derived CAR cells are similarly rapidly-dividing, short-lived and preferentially localized to peri-arteriolar surfaces.

Of note, Fgfr3⁺ fetal chondrocytes of the cartilage template appear to generate osteoblasts and stromal cells much more efficiently than PTHrP⁺ resting chondrocytes of the postnatal growth plate[41]. We postulate that substantial biological differences lie between stem cells in the cartilage template and those in the postnatal growth plate, which make their eventual cell fates fundamentally different. The mechanism conferring Fgfr3⁺ fetal chondrocytes the far superior capability to transform into osteoprogenitor cells is unclear at this stage, representing an important agenda for future investigation.

Dlx5⁺ perichondrial cells are distinct from previously described Osx-expressing osteogenic perichondrial cells that are located in the inner layer of the fetal perichondrium[4]. Dlx5⁺ cells represent an earlier cell population of Osx⁺ cells in the fetal perichondrium, as Dlx5⁺ cell

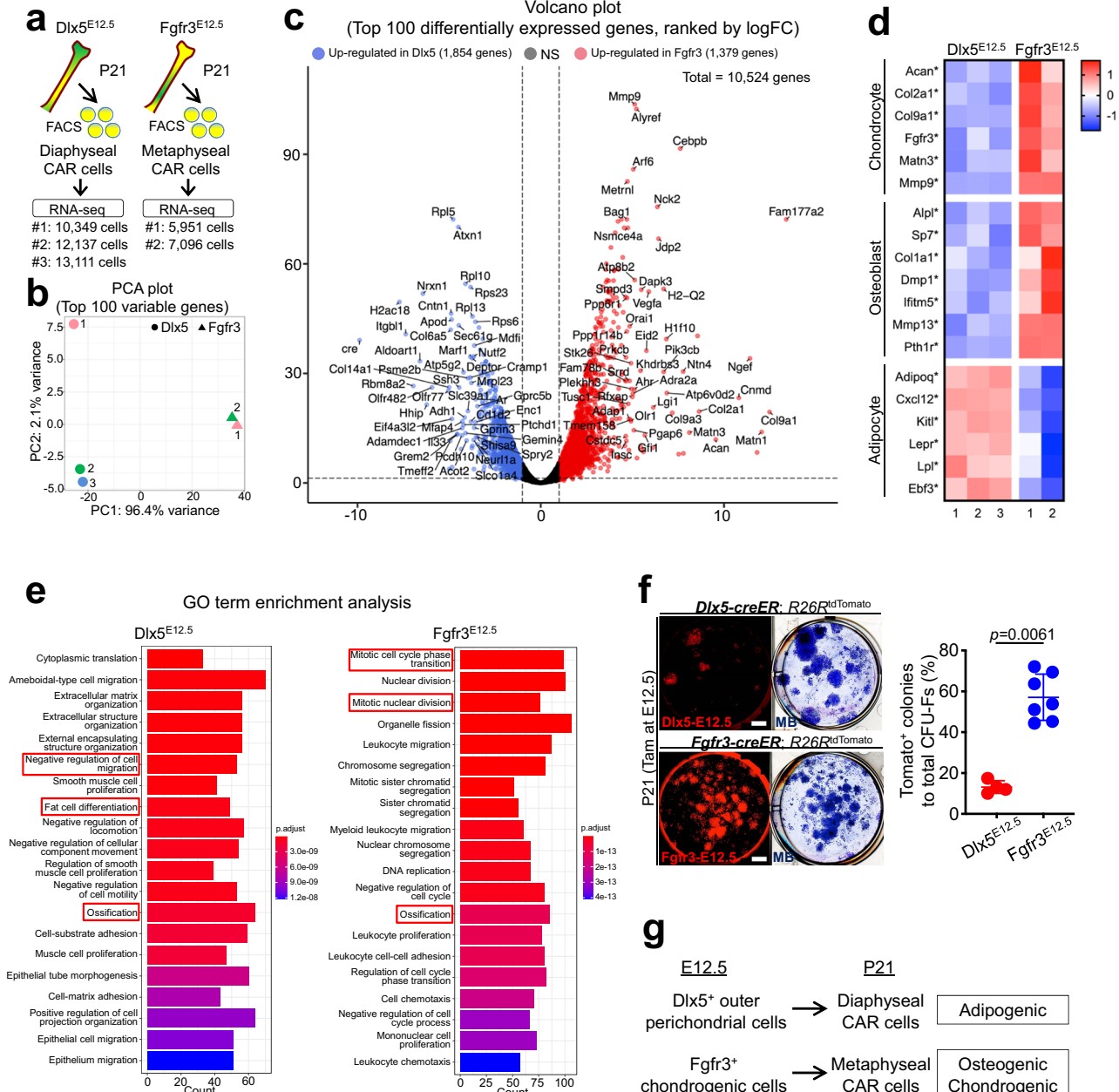

**Fig. 6 | Adipocyte-biased state of Dlx5-creER⁺ perichondrial cell-derived marrow stromal cells. a** Diagram of comparative bulk RNA-seq analysis of Cxcl12-GFP^high stromal cells at P21 that are derived from fetal Dlx5⁺ perichondrial cells and Fgfr3⁺ chondrocytes. GFP^high tdTomato⁺ (Dlx5^CE-E12.5 and Fgfr3^CE-E12.5) cells were isolated from P21 *Cxcl12^GFP/+*; *Dlx5-creER*; *R26R^tdTomato* and *Cxcl12^GFP/+*; *Fgfr3-creER*; *R26R^tdTomato* femurs (pulsed at E12.5) by FACS. n = 3 (Dlx5^CE-E12.5: pooled from n = 4 mice), n = 2 (Fgfr3^CE-E12.5: pooled from n = 4 mice) biologically independent samples. **b** Principal component analysis (PCA) plot of five biologically independent samples. Circles: Cxcl12-GFP^high Dlx5-E12.5 cells, Triangles: Cxcl12-GFP^high Fgfr3-E12.5 cells. *x*-axis: PC1, 96.4% variance, *y*-axis: PC2, 2.1% variance. **c** Volcano plot for differentially expressed genes (DEGs). *x*-axis: log2fold-change, *y*-axis: -log10adjusted *p*-value. Total of 10,524 genes (excluding known pseudogenes). 1379 and 1854 genes upregulated in Dlx5^CE-E12.5 (blue) and Fgfr3^CE-E12.5 (red) cells, respectively. Two-sided exact test for difference in means of the negative binomial distribution, followed by Benjamini-Hochberg (B-H) method adjusted for multiple

comparisons. **d** Heatmap of representative DEGs associated with chondrocyte, osteoblast or adipocyte differentiation. Star: statistically significant DEGs between Dlx5^CE-E12.5 and Fgfr3^CE-E12.5. **e** Gene Ontology enrichment analysis based on 1379 and 1854 genes upregulated in Dlx5^CE-E12.5 and Fgfr3^CE-E12.5 cells, respectively. Top 20 GO terms ranked by B-H adjusted *p*-value. *x*-axis: number of genes enriched in the corresponding GO term. Color indicates adjusted *p*-value. One-sided Fisher's exact test, followed by B-H adjustment for multiple testing. **f** Colony-forming unit fibroblast (CFU-F) assay of E12.5-pulsed *Dlx5-creER*; *R26R^tdTomato* (upper left panels) and *Fgfr3-creER*; *R26R^tdTomato* (lower left panels) bone marrow cells. MB: Methylene blue staining. Scale bar: 5 mm. Right panel: Percentage of tdTomato⁺ colonies among total CFU-Fs. Dlx5^CE-E12.5: n = 4, Fgfr3^CE-E12.5: n = 7 mice. Two-tailed, Mann–Whitney's U-test. Data are presented as mean ± s.d. Exact P value is indicated in the figures. **g** Dlx5⁺ fetal outer perichondrial cells contribute to diaphyseal adipogenic marrow stromal cells, while Fgfr3⁺ fetal chondrocytes contribute to osteogenic metaphyseal stromal cells. Source data are provided as a Source Data file.

---

derivatives sustainably contribute to cortical bone and marrow stromal compartments in postnatal bones, unlike Osx⁺ cell derivatives that contribute only transiently to these fractions. A potential limitation of our tamoxifen-inducible *creER*-based lineage-tracing approaches is

that additional new recombination may occur for several days after tamoxifen injection, while cells are actively translocating to the nascent marrow space. It is therefore technically challenging to formally distinguish the cell fates of a cell population and its immediate

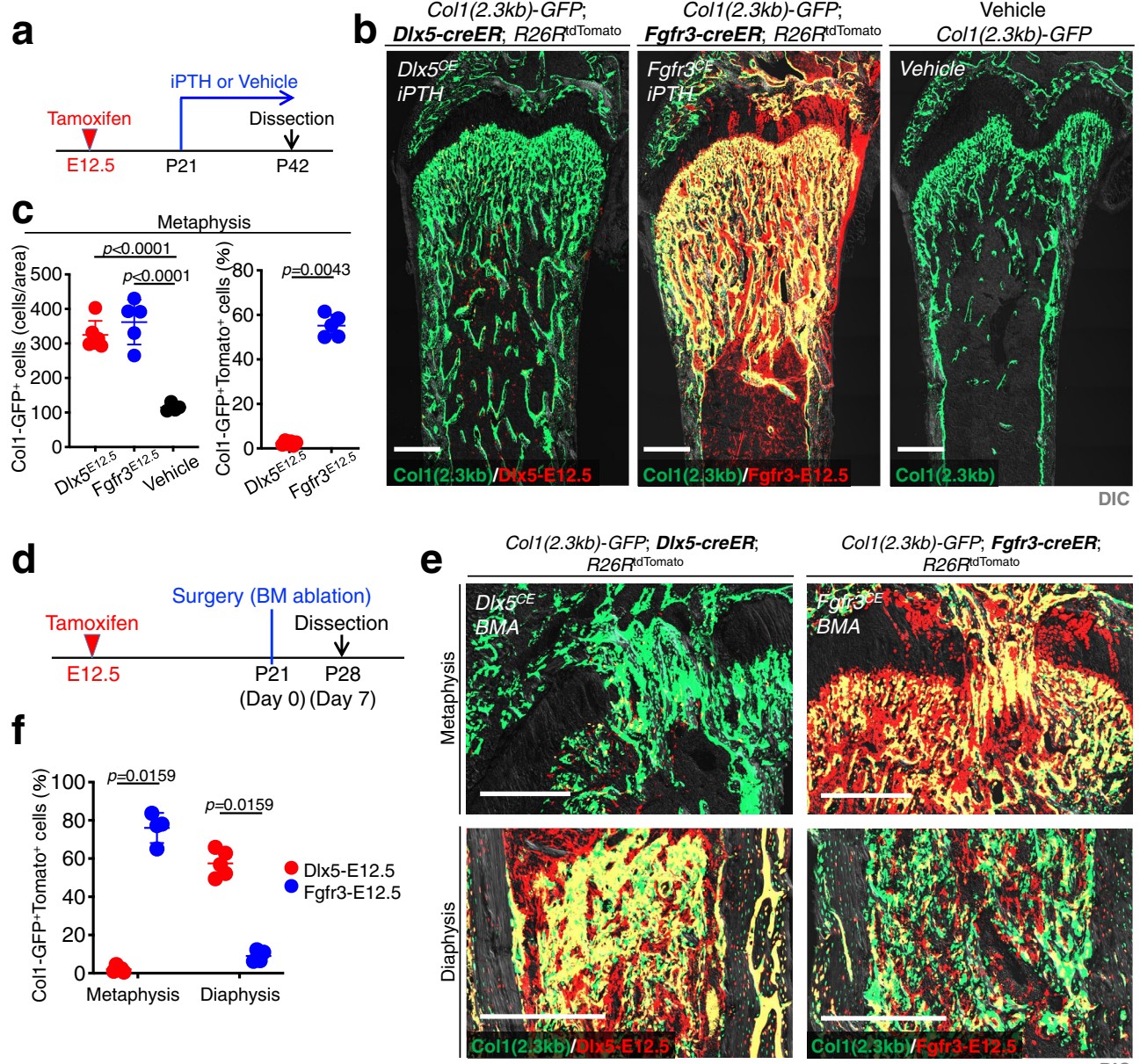

**Fig. 7 | Anabolic and injury responses of Dlx5-creER⁺ perichondrial cell-derived marrow stromal cells. a–c** Differential responses of marrow stromal cells derived from Dlx5⁺ perichondrial cells and Fgfr3⁺ chondrocytes to intermittent administration of parathyroid hormone (PTH) (iPTH). (b): *Col1a1(2.3 kb)-GFP*; *Dlx5-creER*; *R26R*tdTomato (Dlx5CE-E12.5, left), *Col1a1(2.3 kb)-GFP*; *Fgfr3-creER*; *R26R*tdTomato (Fgfr3CE-E12.5, center), and *Col1a1(2.3 kb)-GFP*; *R26R*tdTomato (Control, right) distal femurs, with iPTH or Vehicle for 3 weeks from P21 to P42. Scale bar: 500 μm. (c): Quantification of Col1a1(2.3 kb)-GFP⁺ cells (left) and Col1a1(2.3 kb)-GFP⁺tdTomato⁺ cells (right) in the metaphysis. *n* = 6 (Dlx5CE-E12.5), *n* = 5 (Fgfr3CE-E12.5), *n* = 4 (Control) mice. Two-tailed, One-way ANOVA followed by Tukey's post-hoc test (left). Two-tailed, Mann-Whitney's *U*-test (right). Data are presented as mean ± s.d. Exact *P* value is indicated in the figures. **d–f** Injury-responsive osteogenesis of marrow stromal cells derived

from fetal perichondrial cells (Dlx5CE-E12.5) and chondrocytes (Fgfr3CE-E12.5) cells. (e): Reactive osteogenesis induced by bone marrow ablation in *Col1a1(2.3 kb)-GFP*; *Dlx5-creER*; *R26R*tdTomato (left panels) and *Fgfr3-creER*; *R26R*tdTomato (right panels) femurs at P28 (pulsed at E12.5). These mice underwent surgery at P21. Metaphyseal (upper) and diaphyseal (lower) marrow space after one week of surgery. Scale bar: 500 μm. (f): Quantification of Col1a1-GFP⁺tdTomato⁺ cells in the metaphyseal and diaphyseal bone marrow. Percentage of Col1a1-GFP⁺tdTomato⁺ osteoblasts within Col1a1-GFP⁺ osteoblasts in the metaphysis (left) and diaphysis (right). *n* = 5 (Dlx5CE-E12.5), *n* = 4 (Fgfr3CE-E12.5) mice. Two-tailed, Mann-Whitney's *U*-test. Data are presented as mean ± s.d. Exact *P* value is indicated in the figures. Source data are provided as a Source Data file.

descendants in a fast-evolving process of early bone development. Despite these limitations, we believe that our findings indicate that there may be a long-living cell population within the fetal perichondrium.

Interestingly, the contribution of Dlx5⁺ cell derivatives to cortical bone and marrow stromal compartments of the metaphyseal marrow space appears to be exhausted at later postnatal stages. We assume that this is due to the cell fate of Dlx5⁺ perichondrial cells that are

primarily destined to become long-lived adipocyte-biased marrow stromal cells, which do not readily convert into short-lived osteoblast-biased marrow stromal cells. We therefore postulate that the Dlx5⁺ cell-derived osteoprogenitor pool in the marrow space is gradually replaced by those derived from other cellular sources such as Fgfr3⁺ cells in the metaphyseal marrow space in postnatal stages. Additionally, Osx is also expressed by an osteoblast-biased subset of CAR cells within the postnatal marrow space; however, these Osx⁺CXCL12⁺

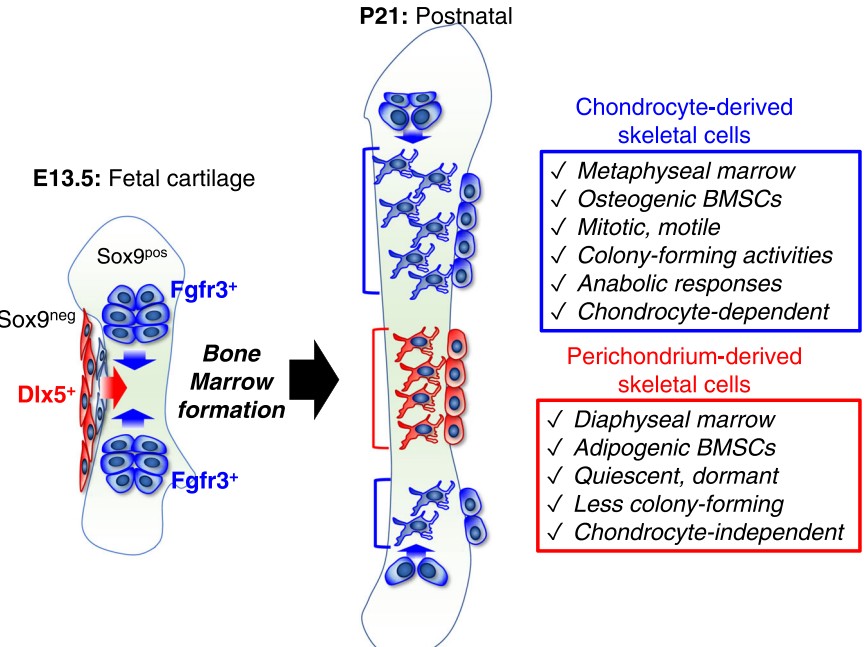

**Fig. 8 | Dlx5⁺ fetal perichondrial cells are important cell-of-origin for distal bone marrow stroma.** Dlx5⁺ fetal outer perichondrial cells translocate into the nascent marrow space and directly differentiate into marrow stromal cells by bypassing a Sox9⁺ state (chondrocyte-independent pathway). These early perichondrial skeletal cells continue to provide osteoblasts and bone marrow stromal cells well into the postnatal stage, unlike Osx⁺ perichondrial cells that have a limited life span. As a result, postnatal bone narrow is characterized by transitional mosaicism composed of both chondrocyte-derived and perichondrium-derived stromal cells.

---

marrow stromal cells are also transient and gradually disappear from this fraction[22], presumably replaced by other cellular sources such as Fgfr3⁺ cells.

Whether these Dlx5⁺ early perichondrial cells possess characteristics as skeletal stem cells, such as those identified in the adult periosteum[42], needs to be determined by further studies. The partnership between the perichondrium and the cartilage template may be one of the major driving forces for mammalian endochondral bone development.

## Methods

### Mouse strains

*Osx-creER*[4] and *Cxcl12^{GFP/+}* [43] mice have been described previously. *Col2a1-cre* (JAX003554), *Col2a1-creER* (JAX006774), *Fgfr3-creER* (JAX025809), *Dlx5-creER* (JAX010705), *Rosa26-CAG-loxP-stop-loxP-tdTomato* (Ai14: *R26R-tdTomato*, JAX007914), *Col1a1(2.3 kb)-GFP* (JAX013134), *Osteocalcin-GFP* (JAX017469), *Osterix-mCherry* (JAX024850), *Ptch1-floxed* (JAX030494), *FVB/NJ* (JAX001800), and *C57BL/6J* (JAX000664) mice were acquired from the Jackson laboratory. *Fgfr3-GFP* (MMRRC: 031901-UCD) mice were acquired from the Mutant Mouse Resource and Research Centers. All procedures were conducted in compliance with the Guidelines for the Care and Use of Laboratory Animals approved by the University of Texas Health Science Center at Houston's Animal Welfare Committee (AWC), protocol AWC-21-0070, and the University of Michigan's Institutional Animal Care and Use Committee (IACUC), protocol 9496. All mice were housed in a specific pathogen-free condition, and analyzed in a mixed background. Mice were housed in static microisolator cages (Allentown Caging, Allentown, NJ). Access to water and food (irradiated LabDiet 5008, Richmond, IN) was ad libitum. Animal rooms were climate controlled to provide temperatures of 22–23 °C, 40–65% of humidity on a 12 h light/dark cycle (lights on at 0600). For all breeding experiments, *creER* transgenes were maintained in male breeders to avoid spontaneous germline recombination. Mice were identified by micro-tattooing or ear tags. Tail biopsies of mice were lysed by a HotShot protocol (incubating the tail sample at 95 °C for 30 min in an alkaline lysis reagent followed by neutralization) and used for PCR-based genotyping (GoTaq Green Master Mix, Promega, and Nexus X2, Eppendorf). Perinatal mice were also genotyped fluorescently (BLS miner's lamp) whenever possible. Mice were euthanized by over-dosage of carbon dioxide or decapitation under inhalation anesthesia in a drop jar (Fluriso, Isoflurane USP, VetOne).

### Tamoxifen and induction of *cre-loxP* recombination

Tamoxifen (Sigma T5648) was mixed with 100% ethanol until completely dissolved. Subsequently, a proper volume of sunflower seed oil (Sigma S5007) was added to the tamoxifen-ethanol mixture and rigorously mixed. The tamoxifen-ethanol-oil mixture was incubated at 60 °C in a chemical hood until the ethanol evaporated completely. The tamoxifen-oil mixture was stored at room temperature until use. Tamoxifen was injected at a dose of 3 mg to pregnant mice intraperitoneally using a 26-1/2-gauge needle (BD309597).

### Intermittent administration of parathyroid hormone

Mice were given once daily subcutaneous injections of PTH (1–34) (100 μg/kg b.w., 4095855, Bachem) for 3 weeks from P21 to P42.

### Bone marrow ablation surgery

Mice were anesthetized through a nose cone, in which 1.5–2% isoflurane was constantly provided with oxygen through a vaporizer. For bone marrow ablation surgery, right femurs were operated, while left femurs were untreated and used as an internal control. After skin incision, cruciate ligaments of the knee were carefully separated using a dental excavator, and a hole was made in the intercondylar region of femurs using a 26-1/2-gauge needle (BD309597). The endodontic instruments (K-file, #35, 40, 45, 50; GC, Tokyo, Japan) were used in a stepwise manner to remove a cylindrical area of the marrow space. The surgical field was irrigated with saline, and the incision line was sutured.

## Histology and immunohistochemistry

Samples were dissected under a stereomicroscope (Nikon SMZ-800), and fixed in 4% paraformaldehyde for a proper period, typically ranging from 3 h to overnight at 4 °C, then decalcified in 15% EDTA for a proper period, typically ranging from 3 h to 14 days. Embryonic samples were not decalcified. Subsequently, samples were cryoprotected in 30% sucrose/PBS solutions and then in 30% sucrose/PBS:OCT (1:1) solutions, each at least overnight at 4 °C. Samples were embedded in an OCT compound (Tissue-Tek, Sakura) under a stereomicroscope and transferred on a sheet of dry ice to solidify the compound. Embedded samples were cryosectioned at 14 μm using a cryostat (Leica CM1850) and adhered to positively charged glass slides (Fisherbrand ColorFrost Plus). Sections were postfixed in 4% paraformaldehyde for 15 min at room temperature. For immunostaining, sections were permeabilized with 0.25% TritonX/TBS for 30 min, blocked with 3% BSA/TBST for 30 min and incubated with rabbit anti-SOX9 polyclonal antibody (1:500, EMD-Millipore, AB5535), rat anti-endomucin (EMCN) monoclonal antibody (1:100, Santa Cruz Biotechnology, sc65495), rabbit anti-MYH3 polyclonal antibody (1:500, Abcam, ab124205), rabbit anti-perilipin A/B polyclonal antibody (1:500, Sigma, P1873) or goat anti-ALPL polyclonal antibody (1:100, R&D, AF2910) overnight at 4 °C, and subsequently with Alexa Fluor 647-conjugated donkey anti-rabbit IgG (A31573) or Alexa Fluor 633-conjugated goat anti-rat IgG (A21049) (1:400, Invitrogen) for 3 hours at room temperature. Sections were further incubated with DAPI (4′,6-diamidino-2-phenylindole, 5 μg/ml, Invitrogen D1306) to stain nuclei prior to imaging.

## RNAscope in situ hybridization

Samples were fixed in 4% paraformaldehyde overnight at 4 °C, and then cryoprotected. Frozen sections at 14 μm were prepared on positively charged glass slides. In situ hybridization was performed with RNAscope Multiplex Fluorescent Detection Kit v2 (Advanced Cell Diagnostics 323100) using *Dlx5* (478151) and *Col2a1* (407221) probes according to the manufacturer's protocol.

## Imaging and cell quantification

Images were captured by an automated inverted fluorescence microscope with a structured illumination system (Zeiss Axio Observer Z1 with ApoTome.2 system) and Zen 2 (blue edition) software. The filter settings used were: FL Filter Set 34 (Ex. 390/22, Em. 460/50 nm), Set 38 HE (Ex. 470/40, Em. 525/50 nm), Set 43 HE (Ex. 550/25, Em. 605/70 nm), Set 50 (Ex. 640/30, Em. 690/50 nm) and Set 63 HE (Ex. 572/25, Em. 629/62 nm). The objectives used were: Fluar 2.5x/0.12, EC Plan-Neofluar 5x/0.16, Plan-Apochromat 10x/0.45, EC Plan-Neofluar 20x/0.50, EC Plan-Neofluar 40x/0.75, Plan-Apochromat 63x/1.40. Images were typically tile-scanned with a motorized stage, Z-stacked and reconstructed by a maximum intensity projection (MIP) function. Differential interference contrast (DIC) was used for objectives higher than 10x. Representative images of at least three independent biological samples are shown in the figures. Quantification of cells on sections was performed using NIH Image J software.

## Cell preparation

For embryonic samples, hind limbs were harvested and incubated with 2 Wunsch units of Liberase TM (Sigma/Roche 5401127001) in 2 ml Ca²⁺, Mg²⁺-free Hank's Balanced Salt Solution (HBSS, Sigma H6648) at 37 °C for 15 min on a shaking incubator (ThermomixerR, Eppendorf). For postnatal samples, soft tissues and epiphyses were carefully removed from dissected femurs. After removing distal epiphyseal growth plates and cutting off proximal ends, femurs were cut roughly and incubated with 2 Wunsch units of Liberase TM and 1 mg of Pronase (Sigma/Roche 10165921001) in 2 ml Ca²⁺, Mg²⁺-free HBSS at 37 °C for 60 min on a shaking incubator. After cell dissociation, cells were mechanically triturated using an 18-gauge

needle with a 1 ml Luer-Lok syringe (BD) and a pestle with a mortar (Coors Tek), and subsequently filtered through a 70 μm cell strainer (BD) into a 50 ml tube on ice to prepare single-cell suspension. These steps were repeated for 5 times, and dissociated cells were collected in the same tube. Cells were pelleted and resuspended in an appropriate medium for subsequent purposes. For cell culture experiments, cells were resuspended in 10 ml culture medium and counted on a hemocytometer.

## Flow cytometry

Dissociated cells were stained by standard protocols with the following antibodies (1:500, eBioscience). eFluor450-conjugated CD31 (390, endothelial/platelet), CD45 (30F-11, hematopoietic), Ter119 (TER-119, erythrocytes), allophycocyanin (APC)-conjugated CD31 (390, endothelial/platelet), CD45 (30F-11, hematopoietic), and Ter119 (TER-119, erythrocytes). Flow cytometry analysis was performed using a four-laser BD LSR Fortessa (Ex. 405/488/561/640 nm) and FACSDiva software. Acquired raw data were further analyzed on FlowJo software (TreeStar). Representative plots of at least four independent biological samples are shown in the figures.

## Single-cell RNA-seq analysis of fluorescence-activated cell sorting-isolated cells

Cell sorting was performed using a four-laser BD FACS Aria III (Ex.407/488/561/640 nm) high-speed cell sorter with a 100 μm nozzle. tdTomato⁺ cells were directly sorted into ice-cold DPBS/1% BSA, pelleted by centrifugation and resuspended in appropriate amount of DPBS/1% BSA (1000 cells/μl). Cell numbers were quantified by Countess II automated Cell Counter (ThermoFisher) before loading onto the Chromium Single Cell 3′ v2 microfluidics chip (10x Genomics Inc., Pleasanton, CA). cDNA libraries were sequenced by Illumina HiSeq 4000 using two lanes and 50 base single-end read, generating a total of ~ 770 million reads, or by NovaSeq 6000 using 150 base pair-end read. The sequencing data was first pre-processed using the 10X Genomics software Cell Ranger. For alignment purposes, we generated and used a custom genome fasta and index file by including the sequences of *tdTomato-WPRE* to the mouse genome (mm10).

## Quality Control and Exploration of scRNA Sequencing Data (Seurat)

We carried out quality control (QC) and cell identification steps, by utilizing Seurat 3.0[44]. We created a Seurat object from the raw gene count matrix, by using the *Read10X()* and *CreateSeuratObject()* functions. Only features detected in at least 10 cells were retained and only cells with at least 500 features were kept for subsequent analysis. In the QC steps, we filtered out low-quality cells. We excluded cells that had: (1) <2000 or >9000 unique feature counts, or (2) >15% mitochondrial content.

To project high-dimensional data into low-dimensional space for visualizing the cluster structure, we applied the functions of *RunUMAP()* with dims = 1:30, *FindNeighbors()* with dims = 1:30, and *FindClusters()* with resolution = 0.5. We removed a total of two clusters, consisting of cells showing little to no tdTomato expression, using *subset()*. Of those, 7889 cells met the QC criteria and were used for downstream analysis.

We explored differentially expressed genes (DEGs) across cell clusters by using *FindMarkers()*. With this approach, we ranked the DEGs with p-values from smallest to largest (statistical significance defined by adjusted p-values <0.05). We used canonical gene markers and the top 100 DEGs of the 10 clusters to assign their cell identities. The UMAP-based cluster visualization and the marker gene expression plots are shown in Fig. 1d, e, respectively. For further analyses of intercellular communication and RNA velocity, we exported the UMAP coordinates of 7889 cells and the clustering results of 10 cell populations (refer to CellChat and scVelo sections).

## Quantitative analysis of cell-cell communication networks of scRNA data (CellChat)

To study how intercellular signaling networks among various cell types shape their differentiation response, we employed CellChat[30]. First, we extracted the normalized count data matrix and cell cluster information from the Seurat object, by applying the function *GetAssayData()* with the settings of assay = "RNA" and slot = "data". Subsequently, we generated a CellChat object with *createCellChat()* and added the corresponding cell identities with *setIdent()*. We applied *subsetData()* to subset the expression data of signaling genes, as recommended by the CellChat protocol.

We used *CellChatDB.mouse*, a database of 2021 validated molecular interactions containing the ligands, receptors, and their cofactors, as the basis for the ligand-receptor analysis. We subsequently applied *identifyOverExpressedGenes()* and *identifyOverExpressedInteractions()* to discover the over-expressed genes as well as the over-expressed ligand-receptor pairs (statistical significance level defined by the Bonferroni corrected *p*-value < 0.05). We utilized *projectData()* to project the gene expression values of ligand-receptor pairs onto protein-protein interaction (PPI) network.

To infer the communication probability of each signaling pathway, we applied *computeCommunProb()*, followed by *computeCommunProbPathway()* and *aggregateNet()* for computing and aggregating communication networks between cells. We used *netAnalysis_computeCentrality()* to compute intercellular communication networks of signaling pathways, thus inferring the role(s) of the 10 cell clusters as the sender, receiver, mediator, and/or influencer.

## RNA velocity analysis (velocyto and scVelo)

We used the sample-specific aligned bam file from Col2a1[cre]-E13.5 mouse (GEO Accession Number: GSM3619209), as the input for velocyto to quantify the unspliced and spliced abundances, generating a loom file[26]. To calculate the RNA velocities of the single cells, we utilized scVelo (version 0.2.2)[45]. We normalized and log-transformed gene expression using the function *scvelo.pp.filter_and_normalize()*, with settings of min_shared_counts = 20, n_top_genes = 5500, flavor = "seurat". We computed the first and second order moments for each cell across its nearest neighbors by using *scvelo.pp.moments()* with settings of method = 'umap', n_neighbors = 30, n_pcs = 30, knn = True. We estimated the velocities by running the likelihood-based dynamical model with *scvelo.tl.velocity(mode = "dynamical")*, and constructed the velocity graph with *scvelo.tl.velocity_graph()*.

To test for the presence of differential kinetics across cell types that could not be well explained by a single model of the overall dynamics, we applied *scvelo.tl.differential_kinetic_test()*. We then recomputed the RNA velocities, by using *scvelo.pp.neighbor()* with similar settings mentioned before, *scvelo.tl.velocity(diff_kinetics = True)*, and *scvelo.tl.velocity_graph()*. To visualize the RNA velocities, we projected velocities onto the UMAP coordinates, using the function *scvelo.tl.velocity_embedding_stream()* with the setting of basis = 'umap'. The UMAP plot is shown in Fig. 1f.

## Single-cell fate mapping (CellRank)

To compute a global map of cellular fate potential, we performed cell fate mapping inference by carrying out the protocol from CellRank[46]. We used *cr.tl.initial_states()* with default settings. We determined the initial cell population to be cluster 5, based on the generalized Perron Cluster Cluster Analysis (G-PCCA) method, implemented in CellRank[46,47]. A representative graph of global probabilistic fates at the single-cell level and the inferred root cell population, is presented in Fig. 1g.

## Bulk RNA-seq analysis of FACS-isolated cells

Dissociated bone marrow cells harvested from P21 littermate mice were pooled based on the genotype [*Cxcl12[GFP/+]*; *Dlx5-creER*; *R26R[tdTomato]*

(pulsed at E12.5, Dlx5-E12.5 mice) or *Cxcl12[GFP/+]*; *Fgfr3-creER*; *R26R[tdTomato]* (pulsed at E12.5, Fgfr3-E12.5 mice)]. Cell sorting was performed using a four-laser BD FACS Aria III (Ex.407/488/561/640 nm) high-speed cell sorter with a 100 μm nozzle. *Cxcl12*-GFP[high]tdTomato[+] cells isolated from Dlx5-E12.5 or Fgfr3-E12.5 mice were directly sorted into ice-cold DPBS/10% FBS and pelleted by centrifugation. Total RNA was isolated using PicoPure RNA Isolation Kit (KIT0204, ThermoFisher), followed by DNA-free DNA removal kit (AM1906, ThermoFisher) to remove contaminating genomic DNA. RNA Integrity Number (RIN) was assessed by Agilent 2100 Bioanalyzer RNA 6000 Pico Kit. Samples with RIN > 8.0 were used for subsequent analyses. Complementary DNAs were prepared by SMART-Seq v4 Ultra Low Input RNA Kit for Sequencing (Clontech 634888). Post-amplification quality control was performed by Agilent TapeStation DNA High Sensitivity D1000 Screen Tape system. DNA libraries were prepared by Nextera XT DNA Library Preparation Kit (Illumina) and submitted for NextGen sequencing (Illumina NovaSeq 6000 150 base for pair-end reads for Dlx5[E12.5] samples [$n = 3$ biological replicates], HiSeq 4000, 50 base for single-end reads for Fgfr3[E12.5] samples [$n = 2$ biological replicates]). Reads files were downloaded and concatenated into a single.fastq file for each sample. The quality of the raw reads data for each sample was checked using FastQC to identify quality problems. Tuxedo Suite software package was subsequently used for alignment (using TopHat and Bowtie2). FastQC was used for a second round of quality control (post-alignment). We employed biomaRt[48] to convert Ensembl gene identifiers into MGI (Mouse Genome Informatics) mm10 gene symbols. To omit known pseudogenes, we only included protein coding genes, defined by their gene biotypes, in addition to reporter genes, such as *cre*, *eGFP*, and *tdTomato*. We utilized edgeR[49] to perform library size normalization and differential expression analysis. For normalization of sequencing depth of each replicate, read counts were scaled to CPM (counts per million). Only genes having CPM greater or equal to 10 in at least two out of the five samples were included. The resulting 10,542 genes were used for subsequent downstream differential expression analysis. We discovered upregulated genes in Dlx5[E12.5] and Fgfr3[E12.5] samples, which were defined by having an absolute number of $\log_2$FC ($\log_2$ Fold Change) greater than 1 and B-H (Benjamini-Hochberg) adjusted *p*-value <0.05. For an unbiased view of the top differentially expressed genes, we used EnhancedVolcanoPlot[50] to construct a volcano plot, as shown in Fig. 6c, highlighting the names of the 100 most differentially expressed genes. The gene ranking was based on descending order by $\log_2$FC. Next, considering all upregulated genes in Dlx5[E12.5] ($n = 1379$) and in Fgfr3[E12.5] ($n = 1854$) cells, we conducted GO (Gene Ontology) enrichment analysis using ClusterProfiler4.0[51]. The top 20 GO enriched biological processes for each condition were ranked by B-H adjusted p-value and were illustrated in Fig. 6e.

## Colony-forming assay and subcloning

Nucleated bone marrow cells were plated into tissue culture 6 well plates (BD Falcon) at a density of <10^5 cells/cm^2, and cultured in low-glucose DMEM with GlutaMAX supplement (Gibco 10567022) and 10% mesenchymal stem cell-qualified FBS (Gibco 12662029) containing penicillin-streptomycin (Sigma P0781) for 10-14 days. Cell cultures were maintained at 37 °C in a 5% $CO_2$ incubator. Representative images of at least three independent biological samples are shown in the figures. For CFU-Fs, cells were fixed with 70% Ethanol for 5 min and stained for 2% methylene blue.

## Statistics and reproducibility

Results are presented as mean values ± S.D. Statistical evaluation was conducted using the Mann-Whitney's *U*-test or one-way ANOVA. A *P* value of <0.05 was considered significant. No statistical method was used to predetermine sample size. Some of the data were excluded from the study because of the pre-established criteria such as problems or failures in identifying correct genotypes or birth dates, and

issues unrelated to the intervention of the study such as spontaneous malnutrition. Sample size was determined on the basis of previous literature and our previous experience to give sufficient standard deviations of the mean so as not to miss a biologically important difference between groups. The experiments were not randomized. All of the available mice of the desired genotypes were used for experiments. The investigators were not blinded during experiments and outcome assessment. One femur from each mouse was arbitrarily chosen for histological analysis. Genotypes were not particularly highlighted during quantification.

### Reporting summary

Further information on research design is available in the Nature Portfolio Reporting Summary linked to this article.

## Data availability

The data generated in this study are provided in the Source Data file. The raw data generated during and/or analyzed during the current study are available from the corresponding author on reasonable request. The single cell and bulk RNA-seq data presented herein have been deposited in the National Center for Biotechnology Information (NCBI)'s Gene Expression Omnibus (GEO), and are accessible through GEO Series accession numbers GSE126966, GSE197933. Source data are provided with this paper. Reference genome mm10 (Ensembl release 93) for transcript annotation is publicly available on Ensembl, including a gtf file [http://ftp.ensembl.org/pub/release-93/gtf/mus_musculus/Mus_musculus.GRCm38.93.gtf.gz] and a fasta file [http://ftp.ensembl.org/pub/release-93/fasta/mus_musculus/dna/Mus_musculus.GRCm38.dna_sm.primary_assembly.fa.gz]. The scRNA-seq data presented in this study is accessible from the following link through the cellxgene platform: https://welchlab.dcmb.med.umich.edu/view/Dlx5/col2e.h5ad/. Source data are provided with this paper.

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

## Acknowledgements

We thank H.M. Kronenberg (Massachusetts General Hospital) for *Osx-creER* mice and T. Nagasawa (Osaka University) for *Cxcl12$^{GFP/+}$* mice. This research was supported by National Institute of Health grants R01DE030630, R01DE026666 (to N.O.), R01DE029181 (to W.O.), R01AR065409, R01AR080654 (to S.W.), R01HG010883 (to J.D.W.) and University of Michigan MCubed 2.0 Grant (to N.O., W.O., and S.W.), MCubed 3.0 Grant (to N.O., W.O., and J.D.W.), JSPS grant JP20KK0356 and JST FOREST Program JPMJFR2111 Japan to Y.M. We thank M. Pihalja and K. Saiya-Cork of the University of Michigan Flow Cytometry Core and T. Lau for supporting this study.

## Author contributions

Y.M. and N.O. conceived the project and wrote the manuscript. Y.M., C.T.A., S.O., M.N., W.O., and N.O. performed the experiment. A.K.Y.C. and J.D.W. performed the bioinformatic analysis. S.Y.W. provided the mice and critiqued the manuscript.

## Competing interests

The authors declare no competing interests.
