## [Peer Review File · Nature Communications]

The fate of early perichondrial cells in developing bonesEditorial Note: This manuscript has been previously reviewed at another journal that is not operating a transparent peer review scheme. This document only contains reviewer comments and rebuttal letters for versions considered at *Nature Communications*.

Reviewers' comments:

Reviewer #1 (Remarks to the Author):

Since the first submission, the authors have performed numerous experiments and revised the manuscript accordingly. The current manuscript gets its strength from providing interesting tools (promotor-driven creER mice) to study the fate of perichondrial cells and chondrocytes during bone development, although there are limitations to the specificity of the described markers as the authors also report in their manuscript, especially with respect to Hes1-creER. To avoid misinterpretation, it should be mentioned in the abstract (and perhaps also title) that the Hes1-creER transgenic mouse model is not suitable to study gene function in the perichondrial cells by targeted gene-deletion, as Hes1-driven cre is also expressed in the surrounding tissues (muscle, endothelial cells).

Minor comment: the last sentence of the abstract is not fully clear: "...perichondrial cells participate in endochondral bone development through a distinct route, by providing a complementary source of skeletal progenitor cells" ; distinct from what, complementary to what

Reviewer #2 (Remarks to the Author):

This manuscript by Matsushita, et al reports the results of single cell sequencing of total early limb bud mesoderm during early condensations and from Col2Cre, then reports on the lineage labeling from various CreERs in the limb bud. The data in the manuscript are not in question, however, the presentation of the data, discussion of the results and the conclusions drawn by the authors are not founded or well-supported.

Single cell analyses of the early limb bud was shown by these authors to result in 7, not well-distinguished clusters. Interesting, and perhaps not surprising given the early differentiation stage. Further examination of limb relevant markers is highly interesting and should not be relegated to supplemental data, but it also shows that the separations in clusters is not clean. Again, not surprising given the early nature of differentiation at this stage. This data is critical, however, as the specificity of the initial expression (and thus, the lineage marked by Cres) impacts the interpretations made by the lineage labeling from these genes.

The next part of the manuscript compares many of these Cres and their resulting lineage. In the first example, Hes1 is examined. Figure 1d-f do not support its expression in the perichondrium as stated by the authors. It is clearly broadly in scattered cells in the lateral plate mesoderm that is not cartilage - this does not define perichondrium. It actually seems to miss almost entirely what one would expect to be the perichondrium just outside the Sox9 cells (as does the Notch reporter). Further, its much broader lineage at 3 days after induction (including a small number of cartilage cells) may not represent lineage labeling, but rather the tdTom becoming fully active. The supplemental data demonstrating lineage labeling to nearly everywhere supports this possibility - again, its relegation to the supplemental data is unacceptable if the authors wish to support specific use of this Cre. Much more detailed time points would have to be examined to support their claims regarding a specific population being labeled and, in any case, it would not be the perichondrium as judged from this data, and it does not appear likely to be perichondrium. Supplemental data 5a also shows a population that one would not describe as perichondrial (although it is not in the cartilage).

The Col2Cre, also cannot be clustered into perichondrium based on Hes1 expression as the authors DO NOT support that Hes1 is perichondrial and the Col2Cre expression is very broad, much too broad to be labeled perichondrium. Also, the genes reported to also be in this population (Dlx, Runx, Prrx, Sps7) are not highly uniform and this information is also in supplemental data. Of note, there have been genes reported to be in perichondrium relatively specifically, including those reported by Bandyopadhyay (Tabin) and Rux (Wellik) have been shown to be in this population - none of these are shown. Critically from this point, the authors move forward from the notion that anything not in the Sox9-expressing zone is perichondrium and this invalidates much of the rest of the conclusions as the starting point is a non-specific population.

The authors then take this a step further and suggest that the cell fates from these and additional Cres (Dlx5, Osx) represent distinct cell populations. This is perhaps the least supported aspect of the study. None of these genes have the initial specificity (by expression, single cell data or early lineage) to support that they represent a specific precursor population. The following analyses and in vitro comparisons gets muddier, with little useful information for those interested in this field.

As with any single cell analyses, each cell expresses 1000s of individual genes, most of which are not be highly specific to the sub-population that cell is clustered into based on all the data - like those interrogated here. In the absence of highly specific gene expression to begin the analyses, the lineage and comparisons are of little scientific value. In this publication, the authors compare several loosely specific (or worse) Cres, and attempt to make conclusions about the 'sub-populations' these Cres lineage label. This information does not provide novel insight, knowledge or meaningful biological comparisons. It certainly does not support the title and main conclusion of the manuscript.

Critically, the two main goals of the paper are stated in the revision to define the cellular diversity of perichondrial cells, chondrocytes and precursors by single-cell seq. They provide some new information on this front, but much more work would be required to make scientifically rigorous conclusions about sub-types potentially identified. The second goal of characterizing novel CreERs for cell-type specific deletions is wholly unsupported. The data does not support any of the Cres represent a meaningful sub-type. To publish it as such will provide a potentially poor premise for future investigations.

REVIEWER COMMENTS

Reviewer #1 (Remarks to the Author):

Overall comments:

There are many comments in the introduction that are surprising given the experience of the contributing author. The first reference (to his own work), states that osteoblasts can be generated from multiple cellular origins; this refers to several published papers, his being the only ones cited, that reports various Cres are capable of lineage labeling into osteoblasts. If it were true that there is only a single origin of osteoblasts, one will recognize that more than one gene would be expressed in those cells and, thus, many Cres would lineage label into osteoblasts. This does not automatically denote subsets, but potential cells states, the stochastic nature of expression, the efficiency of Cres and induction of these Cres, etc.

Likewise, it is remarkable that the authors states that the identity of the perichondrial cells that give rise to *Osx*-expressing cells and osteoblasts is unknown. There are several publications that have demonstrated that *Hox*-expressing cells are such a population. Not only do these cells mark the early outer perichondrial layer and give rise to osteoblasts, but lineage labeling also shows that lineage cells are maintained in bone marrow stromal cells as well as other stromal progenitors and permanently mark stromal progenitor populations. It is remiss that this senior author is apparently unaware of them, nor are any cited. This also raises significant concerns about the premise of the contribution and, thus, its impact. The authors errantly state this work in context of its originality of being the first perichondrial Cre.

Overall, there are perceived over-interpretations of data, built primarily on this author's previous work. There is concern that over-interpretation of data leads to inaccuracies in our understanding of the field. The authors are encouraged to be inclusive of all literature that relates to this field and place these findings in an appropriate broader context.

Unrelated to the above comments, one questions whether the field continues to be moved forward by continued complex interpretations of Cre lineages and downstream cells. The majority of this manuscript reports new Cre lineages that are interesting, but not easily interpretable. Thousands of genes are going to be expressed in each and every sub-population that exists, including perichondrial cells. Their usefulness could be determined by using them to address a biological question that might be dissected. The authors begin this potentially in Figure 7 (Pth), but this one figure on biology conducted at time points far removed from the lineage reporting at much earlier stages is premature.

Comments on manuscript data and discussion of this data:

The nine clusters in Figure 1e are poorly defined. There is remarkably little separation distinguished and the populations could be better annotated. Potential additional expression that could be included in 1e is *ColX*, *osteopontin*, *osteocalcin*, other genes that define the rest of the groups. The most useful papers that contain sc-seq data include an interface for searching in the supplemental materials. A moderately accomplished biostatistician can load data in one of these searchable formats and this should be recommended for all publications that include single cell data.

Whether 55% of a population represents a large fraction (stated percent of undifferentiated perichondrial cells) is a questionable descriptor (Fig 2). Perhaps 'half' should be used? It is unclear in the text what metric was used for the denominator. That only half of cells are labeled raises questions about the usefulness of the Cre – labeling and using it as such might be misleading and lead to confounding results. Moving forward with the assumption that you are following perichondrial cells when only half are labeled is a concern. The premise of the conclusions drawn are strongly impacted by this concern.

**That ~20% of the *Osx*-CE cells label chondrocytes – after only 24 hours of lineage labeling –

presents significant challenges in the claim of separation of populations and labeling them as osteogenic perichondrial cells is a bit muddy.

Figure 3 reports highly interesting data on cells moving into the bone marrow space. The authors note the high contribution from both *Osx*-CE and *Fgfr3*-CE, but the data seems to indicate much higher contribution from *Osx*CE. This is not easily consistent with the idea that early bone marrow space is from the cartilage template – it seems that *Fgfr3*-Cre contribution should be higher? Further, with labeling at E12.5, much of the cartilage template is unlabeled as it is established prior to E12.5 (as supported in Figure 2). While very interesting data, the interpretation of this data is less clear. In Fig 3b, how did the *Dlx5* lineage come to the center of the trabecular area (perhaps more time points between E12.5 and E15.5 are needed)? This data suggests the Groove of Ranvier cells travel laterally and is inconsistent with cells from this central region being chondrocyte-derived. It would be interesting to note the time points from E12.5 to E18.5 that permits a visualization of the presumed migratory path of these cells. It is also intriguing that there are many *Osx*- and *Fgfr3*-lineage labeled cells in this region that are not marked with *Col1*. It seems there is an opportunity here, by understanding overlap, how these are related in this area. While the authors denote comparable percentages between *Dlx*- and *Fgfr3*-Cres, *Osx*-Cre is not discussed. With the amount of overlap in other areas noted and *Dlx* only labeling 50% of perichondrial cells, the conclusion that trabecular osteoblasts derive 50/50 from perichondrial and chondrocyte sources seems over-interpreted (and potentially internally inconsistent). Further, the absolute number of cells contributed by each Cre lineage does not allow a comparison, particularly given the differential efficiencies and/or limitation of the Cres and the inability to assess a total population (part of Fig 3c) - this does not lend to a meaningful use of total numbers to reflect a direct comparison.

Figure 4g is a strikingly interesting result and deserves significantly more attention (for which room could be made by moving some of the earlier panels to supplemental). Here, the phenotype in *Fgfr3*-*Ptc1* mutants (and *Dlx*-*Ptc1*) is worthy of further investigation and incomplete as presented. It is difficult to understand how bm space can be reduced and cartilage length not increased with total lengths not being different. For each N, it may be more appropriate to normalize bone marrow length/total length to ensure this is not a rigor issue with Ns being too low to cause significantly lower bone length but having barely significant changes in bm length. It seems there is a missing piece of information here that might require more Ns or normalized measurements. For both mutants, assessing the changes in trabecular length, length of differentiating cartilage columns and other measures that further define what the Hh GOF is causing is critical to understanding the phenotype; this phenotype is under-assessed. Again, the interpretation that *Dlx* cells regulate bone marrow formation without the chondrocytes is a very liberal 'suggestion' from this panel of information. Much more could be learned here without significant extra effort on the sections from these animals.

Fig 5 shows a relatively expected difference of contribution to bm reticular cells. However, there are two important things this data reveals that the authors do not comment on and is critical for the overall interpretation of their data. The first is the apparent relatively low contribution to osteocytes. This is not quantified (but could be). The data shown shows surprisingly little contribution from either Cre – not consistent with having identified the two sources of osteoblasts and reticular cells. This is in contrast to the report using *Hoxa11*Cre which shows complete labeling of osteoblasts and osteocytes where the perichondrium is labeled (Pineault, 2019; Song, 2020). Perhaps this is due to only 50% of the perichondrium being labeled with this Cre? Related, the endosteum, a source of osteoblasts and the majority of cortical bone growth as shown by several recent publications, remain strikingly unlabeled, perhaps accounting for this 'lost' pool of progenitors. The authors' report that 19% of the reticular cells are from *Dlx*-Cre lineage and 26% from the *Fgfr*-Cre lineage. This is less than 50% of total? Drawing conclusions on discovering the two sources of reticular cells without accounting for missing information (the other 60% of the *Cxcl12*+ cells) should be addressed or clarified.

While it may have been published that *Col2a1*-Cre is a chondrocyte subset, the marking of

perichondrial cells suggest it is a more general marker of cells that express collagen 2 – connective tissue. This author agrees that Col2a1 is not a particularly informative lineage marker, but this data and discussion can probably be skipped as it does not add value.

The bulk-seq data reported in Figure 6 is interesting. The gene expression shown in Fig 6c is also interesting. As changes are small (not surprising), it seems relevant to show these genes in some context of what biggest changes are between the two populations in addition to the ones the authors picked out to highlight. Some presentation of the more complete and unbiased assessment of the data needs to be shown and discussed in the context of the gene expression changes they chose to highlight.

The CFU-F assay goes back to looking at only Fgfr3-Cre and Dlx-Cre lineage bm cells – how many of these are Cxcl12-positive in each set? Was this stated? The relationship to bigger story seems parenthetical. Certainly a paragraph following this data to state that the theory presented in the previous paragraph is supported could be removed.

Figure 7 should be removed in its entirety. The focus of this study on Pth is peripheral and the later time points of investigation, well past when the two Cres have separated almost entirely spatially, make for a messy, unrelated, but perhaps beginning of an interesting story discussing what genetic perturbation of half of the perichondrium and most of the cartilage causes. It is premature as presented currently.

Discussion:

The work shared here does not provide a cell of origin for the periosteum as stated. The DlxCre does mark some of the perichondrium, at a stage where perichondrial cells are already present. It does not speak to their origin; this may have been misstated.

Reviewer #2 (Remarks to the Author):

In this manuscript, Matsushita et al. examine Dlx5 versus Fgfr3-lineage cells using a creERT based lineage tracing approach as a means to examine the eventual fates and roles of cells in the early developmental perichondrium versus the developmental cartilage template. Taken together, this manuscript addresses a novel area of the literature in examining the function and fates of the early perichondrium, and identifies several important distinctions between the derivatives of the perichondrial cells versus other skeletal populations. Perhaps chief among these and the most interesting finding reported here is that this data suggests distinct developmental origins for the recently described osteogenic and adipogenic subsets of Cxcl12 abundant reticular cells (CAR cells). This is particularly important as one of the central questions regarding these two cell types is whether they represent the recent bifurcation of a single progenitor present in adults that gives rise to both cell types or whether these are two wholly distinct cell lineages with distinct origins that happen to share some markers in common. The data presented here argues for perhaps the more surprising and interesting latter possibility. Additionally, while there are areas that require qualification or clarification as indicated below, the methods used are robust and appropriate. Taken together, there is overall enthusiasm for the importance and novelty of this study.

Major point

1. It is an interesting, and likely very important, finding that dlx5 and fgfr3 account for distinct groups of CAR cells that appear by expression profile to correlate with adipo-CAR and osteo-CAR populations. However, this finding would be greatly strengthened if it was demonstrated to correlate with an eventual maturation of the dlx5-lineage putative adipoCAR cells into marrow adipocytes, and conversely maturation of the FGFR3-lineage putative osteo-CAR cells into marrow/trabecular osteoblasts with extended chase periods. Can these groupings be observed to correlate with the

osteolectin-based classification of CAR subsets proposed in Shen et al. Nature 2021? In general, comparison to the Shen et al. paper as well as deeper comparison to the Baccin et al Nat Cell Biol 2020 paper is warranted. The Shen et al. report should be cited.

2. An important limitation of using this creERT-based lineage tracing approach to assess cellular translocation, especially over the fairly fast progression of early murine skeletal development, is that it is difficult to exclude that there is additional new recombination occurring up to several days after the initial tamoxifen injection in the populations apparently translocating. This is especially a concern given the demonstrated long half-life of tamoxifen. It doesn't appear that any of the data presented strongly excludes this possibility. It is appreciated that it is difficult to formally exclude this possibility, so it would be sufficient to acknowledge this limitation prominently.

Minor points and discussion:

1. In the Fig 1 single cell seq study, how is it determined which clusters represent chondrocytes and which clusters represent perichondrial cells? As the authors are aware, the expression of classically chondrocyte associated transcripts in perichondrial cells make this distinction not straightforward on the basis of transcriptional data alone.

2. Previously the authors in the Ono et al. Nat Cell Biol 2014 had used studies of a Sox9-creERT to generate data supporting the conclusion that long-lived periosteal/perichondrial progenitors/stem cells express Sox9. How do the authors interpret the apparent exclusion of Sox9 from the Dlx5 population in light of this prior data? Is Sox9 still a marker of skeletal stem cells in this context? It is appreciated that the setting being studied here is different than that in earlier studies and that no formal claims are made here about the Dlx5-lineage cells including a stem cell, nevertheless this finding was unexpected.

3. Is the low numbers of DLX5 lineage cells that become col1 expressing osteoblasts just a factor of the timepoint chosen or is this more reflective of an intrinsically low rate of these cells contributing to the osteoblast pool?

4. The wording on line 152 implies that the osx-lineage cells in the perichondrium are moving into the marrow space. However, it would appear from the data to be just as likely, or perhaps more likely, that the osx-lineage cells populating the marrow are derived from osx-expressing cells within the cartilage template, not the perichondrium. It is suggested that this wording be adjusted to avoid indicating translocation or at least to indicate the uncertainty regarding this.

5. Caution is advised regarding the use of the term "undifferentiated" with reference to Fig 2. The Col1a1-negative DLX5-lineage cells may be partially differentiated or differentiated to a non-osteoblast fate. Similarly, it is recommended that a more plainly descriptive term (e.g, col1a1-negative) be used instead of undifferentiated throughout.

6. When the authors use the term "cortical osteoblasts" such as in line 170, is that used as a synonym for "periosteal osteoblasts" or "periosteal and endosteal osteoblasts"? It would be helpful to be explicit about which sides of the cortex are being labeled.

7. Optionally, it may be easier to follow the discussion starting in line 207 if annotations are used to refer to the cluster (e.g., Dlx5 and Fgfr3 clusters) names rather than the numbers assigned.

8. In the Ono et al. Nat Cell Biol 2014 report, the Osx-creERT tracing did label CXCL12 abundant reticular cells. Is lack of similar labeling noted in line 250 just due to differences in timing between the present study and the author's past work? If so, that might be a helpful point to cover in the discussion.

9. It would be helpful for the authors to compare the labeling seen with FGFR3 versus their prior work with PTHrP-creERT. Are differences in the marrow stromal populations labeled due to technical differences in these two cre lines or biologic differences in the labeled populations? Underlying this question is a desire to understand the parallels between the cellular composition and eventual fates of the initial cartilage template versus the later growth plate resident stem cells. Given the complexity of the literature on skeletal cell types, in general expansion of the discussion to more strongly tie the current results to past studies would strengthen the manuscript.

10. The description of the scRNA-seq analysis is generally good, however it would be helpful to clarify whether cell cycle state was regressed out and how cell cycle state parsed out among clusters. It

would also be helpful to visualize the number of UMIs or unique genes per cell to see if sampling issues are driving any of the clustering.

11. The statement "Our findings therefore indicate that there is a self-renewing cell population within the fetal perichondrium" may not be justified by the data. No formal self-renewal studies were performed, and no data is provided to show a specific population of Dlx5-lineage cells that retains their defining characteristics over time, such as through serial transplantation. Additionally, the timing of chase periods analyzed here is fairly short. This statement should be softened.

Reviewer #3 (Remarks to the Author):

In this paper, Matsushita et al perform lineage tracing studies on bone formation in mice using two novel marker genes identified through an initial exploration of an scRNA-seq dataset. Through complex but beautifully designed experiments the authors make use of conditional reporters and knockouts to show the differences in contributions from Dlx5+ and Fgfr3+ cells in formation and development of bone marrow. Further, again building on a theory developed from the scRNA-seq analysis, the authors investigate the role of hedgehog signaling between the involved cell types.

Ideally more biological replicates would have been used, but there are no substantial issues with this manuscript. At times the logic for the results from the lineage tracing experiments can be hard to follow due to the many factors involved (multiple conditional reporters, multiple locations, etc). This is inherent in the complexity of the system studied, but perhaps simple illustrations could be used to summarize the conclusions for each figure?

There are a few minor presentation issues:

Panels showing gene expression have no units: Fig. 1e, Fig. 4e,f. Are these counts of UMIs?

In the text, the sub-panels of Fig.3c are referred to as 'leftmost panel', 'left center panel', 'center panel', 'right center panel', and 'rightmost panel'. It would be more clear to refer to these as 'leftmost', 'second from the left', 'third from the left', 'fourth from the left', and 'rightmost panel'.

The text refers to 'Fig.4 red dotted area', but it is not clear what this refers to in the figure.

Also in Fig. 4a, what unit is 'interaction strength'?

Is there a reference for the fact that deletion of Ptch1 leads to constitutive activation of Hedgehog signaling?

In the methods section it says cell numbers were quantified by a 'Countless II' instrument. The correct name is 'Countess II'.

Response to Reviewers: NCOMMS-20-02880A-Z

We would like to thank the reviewers for their careful reading and constructive comments on our manuscript. We have performed additional experiments as requested by the reviewers, and substantially improved the manuscript to address the reviewers' concerns. Please see below in our point-to-point responses to the reviewers' critiques.

Response to Reviewer #1

Thank you very much for your constructive comments and critiques.

1. <Reviewer>

Overall comments:

There are many comments in the introduction that are surprising given the experience of the contributing author. The first reference (to his own work), states that osteoblasts can be generated from multiple cellular origins; this refers to several published papers, his being the only ones cited, that reports various *Cres* are capable of lineage labeling into osteoblasts.

<Response> (##2)

Thank you very much for pointing out this important deficiency of the manuscript, and we apologize for the oversight. We have now cited the original articles in the revised introduction, including *LepR⁺* stromal cells (Zhou et al., *Cell Stem Cell*, 2014), *Col10a1⁺* hypertrophic chondrocytes (Yang et al. *PNAS*, 2014), *Osx⁺* cells (Maes et al. *Dev Cell*, 2010), *aSMA⁺* periosteal cells (Matthews et al. *JBMR*, 2014), fracture callus chondrocytes (Zhou et al. *PLoS Genet*, 2014), *Gli1⁺* progenitor cells (Shi et al. *Nat Commun*, 2017).

2. <Reviewer>

If it were true that there is only a single origin of osteoblasts, one will recognize that more than one gene would be expressed in those cells and, thus, many *Cres* would lineage label into osteoblasts. This does not automatically denote subsets, but potential cells states, the stochastic nature of expression, the efficiency of *Cres* and induction of these *Cres*, etc.

<Response>

We would like to emphasize that our arguments on multiple origins of osteoblasts specifically refer to those located in different anatomical locations of bones, sometimes across different stages of development – for example, perichondrium, cartilage template, growth plate, periosteum, endosteum, as well as metaphyseal bone marrow and diaphyseal bone marrow. We do not intend to refer to cells at the same location but with different cell states, which we believe that the reviewer mentions above.

Our argument relies on the availability of a *creER* line that can specifically mark a group of cells at a specific location (for example, *Pthrp-creER* for resting zone chondrocytes, *aSMA-creER* for periosteal cells). If these cells lineage-trace into osteoblasts, we would be able to draw a conclusion on potential anatomical origins of osteoblasts. If the findings of multiple cell type-specific *creER* lines are combined, we would be

able to conclude that multiple cellular sources (e.g. resting chondrocytes and periosteal cells) can independently generate osteoblasts, leading to the concept that there are multiple origins of osteoblasts.

In contrast, it would be difficult (almost impossible) to determine whether cells marked by different cre/creER lines at the same location (for example, *Lepr-cre* and *Cxcl12-creER* for central marrow stroma) represent distinct cellular sources of osteoblasts. Without spatial information, we cannot distinguish whether the differences in their cellular behaviors are dictated by potential fluctuations of cell states, the stochastic nature of *cre* expression or the efficiency of *cre* induction, as the reviewer pointed out. Therefore, we would like to emphasize that the initial location of the cells is the most important information in interpreting the lineage-tracing results.

We demonstrate in our study that the two *creER* lines (*Dlx5-creER* and *Fgfr3-creER*) mark cells in different compartment of developing bones (perichondrium and cartilage template, respectively). We understand that there are still many technical caveats to be carefully considered to correctly interpret the origins of osteoblasts. However, due to their well-defined anatomical locations, we believe our results are unaffected by potential cell states or issues related to *cre* expression and induction.

3. <Reviewer>

Likewise, it is remarkable that the authors states that the identity of the perichondrial cells that give rise to *Osx*-expressing cells and osteoblasts is unknown. There are several publications that have demonstrated that *Hox*-expressing cells are such a population. Not only do these cells mark the early outer perichondrial layer and give rise to osteoblasts, but lineage labeling also shows that lineage cells are maintained in bone marrow stromal cells as well as other stromal progenitors and permanently mark stromal progenitor populations. It is remiss that this senior author is apparently unaware of them, nor are any cited. This also raises significant concerns about the premise of the contribution and, thus, its impact. The authors errantly state this work in context of its originality of being the first perichondrial Cre.

<Response>

The reviewer's points are well taken. However, we would like to respectfully point out that the elegant *Hoxa11* study that the reviewer mentioned (Pineault et al. Nat Commun 2019) is only pertinent to the zeugopod bones (i.e. ulna and radius in the upper extremity, and tibia and fibula in the lower extremity), because *Hoxa11* is expressed solely in the zeugopod. Obviously, *Hoxa11*-expressing cells do not exist in other bones, including the femur that we studied in our current study. The counterpart of *Hoxa11*⁺ cells in other bones has not been reported by these authors.

In addition, we would also like to respectfully point out that these authors described that *Hoxa11* is expressed not only by cells in the perichondrium, but also by chondrocytes of the cartilage template at E14.5, as assessed by *Hoxa11-GFP* and *Hoxa11-creER; R26R-tdTomato* (please refer to Figure 3a-c of Pineault et al. Nat Commun, 2019, 10:3168 as shown below). Therefore, in the zeugopod limb, *Hoxa11*⁺ cells appear to include both *Dlx5*⁺ perichondrial cells and *Fgfr3*⁺ chondrocytes in the cartilage template.

Figure R1-1

Figure 3a-f of Pineault et al. Nat Commun, 2019, 10:3168

For the reasons stated above, we believe that the Hoxa11 study by Pineault et al. does not compromise the novelty of our work. Therefore, we maintain our standpoint that *Dlx5-creER* represents a novel cell-type specific line for early fetal perichondrial cells.

(##2) However, considering the importance of these works that can be complimentary and synergistic with our work, we cited these two-relevant works (Pineault et al. Nat Commun, 2019, Song et al. PNAS, 2021) in the revised introduction.

4. <Reviewer>

Overall, there are perceived over-interpretations of data, built primarily on this author's previous work. There is concern that over-interpretation of data leads to inaccuracies in our understanding of the field. The authors are encouraged to be inclusive of all literature that relates to this field and place these findings in an appropriate broader context.

<Response> (##2)

We appreciate the reviewer's critical note. In the revised manuscript, we updated our literature relevant to this topic and placed our work more comprehensively in the context of published findings. We added the following new sentences in the introduction for this purpose.

The following paragraph was added to the Introduction:

Line 57-66:

Previous studies reported several constitutively active cre and inducible creER lines that can target perichondrial cells, including Prrx1-creER¹⁴, Hoxa11-creER^{15,16}, Sox9-creER¹⁷, Axin2-creER¹⁸, Ctsk-cre¹⁹ and Fsp1-cre^{20,21}. While these studies substantially contributed to the field, these reported lines may have potential limitations. In particular, these creER lines also mark at least some of chondrocytes within the growth plate. Moreover, Hoxa11-creER is active exclusively in the zeugopod bones (ulna, radius, tibia and fibula)¹⁵, while Sox9-creER is specific to perichondrial cells exclusively in the rib cartilage¹⁷. Additionally, the constitutively active cre lines lack temporal control and have activities in other cell types, such as osteoclasts (Ctsk-cre) and other fibroblasts (Fsp1-cre). Therefore, a perichondrial cell-specific inducible mouse genetic tool would facilitate the understanding of the function of early perichondrial cells.

5. <Reviewer>

Unrelated to the above comments, one questions whether the field continues to be moved forward by continued complex interpretations of Cre lineages and downstream cells. The majority of this manuscript reports new Cre lineages that are interesting, but not easily interpretable. Thousands of genes are going to be expressed in each and every sub-population that exists, including perichondrial cells.

<Response>

The reviewer's point is well taken. We would like to point out that the analysis of a cell lineage requires only one well-verified, tractable marker that can accurately capture a defined group of cells at the specific time of tissue development. This one particular "marker" gene should be the representative of the metagene (a set of genes) that collectively reflect a specific cell state of a specific cluster of cells. It is within our expectation that the defined group of cells express thousands of other genes which may overlap with other cell types.

For example, we demonstrated in Figure 1 that fetal perichondrial cells express a broad set of genes that are also expressed by fetal chondrocytes. However, there are a small subset of genes that make it possible for us to distinguish the two cell types. *Dlx5* and *Fgfr3* were identified as an example of these genes, as we demonstrate in the manuscript. The novelty of our work is that we capitalize on these two novel markers and deconvolute the cell lineage in vivo, which cannot be done with static transcriptomic approaches including single-cell RNA-seq analyses.

6. <Reviewer>

Their usefulness could be determined by using them to address a biological question that might be dissected. The authors begin this potentially in Figure 7 (Pth), but this one figure on biology conducted at time points far removed from the lineage reporting at much earlier stages is premature.

<Response>

We would like to point out that we also performed the functional experiments to interrogate their roles at a much earlier time point (E18.5) in the context of constitutively activated Hedgehog signaling (**Figure 4g**). This functional study demonstrates that both *Dlx5*⁺ and *Fgfr3*⁺ cells are important for regulating the formation of the marrow space within the cartilage template, showing the utility of the two *creER* lines that we identified in this study to interrogate the biological significance of these particular cell types.

7. <Reviewer>

Comments on manuscript data and discussion of this data:

The nine clusters in Figure 1e are poorly defined. There is remarkably little separation distinguished and the populations could be better annotated. Potential additional expression that could be included in 1e is *ColX*, osteopontin, osteocalcin, other genes that define the rest of the groups.

<Response>

We agree with the reviewer's assessment that there is remarkably little separation between the nine clusters that we identified, supporting our statement, "*The fetal chondrocyte-perichondrial cell lineage was largely contiguous and shared an overlapping set of marker genes*" (Line 94-95).

We also included the feature plots of *Col10a1* (encoding ColX), *Spp1* (encoding osteopontin) and *Bglap* (encoding osteocalcin) in the Supplemental Figure, as requested by the reviewer. Importantly, these genes did not appear to be expressed at a high level at this stage (E13.5), reflecting the fact that chondrocytes at the center of the cartilage template have not yet undergone complete hypertrophy, and inner perichondrial cells adjacent to prehypertrophic chondrocytes have not yet undergone terminal osteoblast differentiation.

Figure R1-2 (new Supplementary Figure 2)

8. <Reviewer>

The most useful papers that contain sc-seq data include an interface for searching in the supplemental materials. A moderately accomplished biostatistician can load data in one of these searchable formats and this should be recommended for all publications that include single cell data.

<Response> (##3)

We have increased accessibility of our scRNA-seq dataset by providing a link to Cellxgene, open source cell visualization tool, as requested by the reviewer. Now the reviewers and potential readers of this study have easy access to this dataset without a skill for coding.

The following sentences were added to the Methods, under Data Availability section:

The scRNA-seq data presented in this study is accessible from the following link through the cellxgene platform: <https://welchlab.dcmf.med.umich.edu/view/Dlx5/col2e.h5ad/>.

9. <Reviewer>

Whether 55% of a population represents a large fraction (stated percent of undifferentiated perichondrial cells) is a questionable descriptor (Fig 2). Perhaps 'half' should be used? It is unclear in the text what metric was used for the denominator.

<Response>

We have now changed the word to “*approximately half*” as suggested. We also clarified the denominator as Col1a1-GFP-negative perichondrial cells in the revised manuscript.

10. <Reviewer>

That only half of cells are labeled raises questions about the usefulness of the Cre – labeling and using it as such might be misleading and lead to confounding results. Moving forward with the assumption that you are following perichondrial cells when only half are labeled is a concern. The premise of the conclusions drawn are strongly impacted by this concern.

<Response>

The reviewer’s point is well-taken. However, we respectfully disagree with the reviewer’s comment. We would like to bring your attention to our data in Figure 2a-1 that *Dlx5-creER* specifically marks a specific layer of the perichondrium (but still outside of the osteogenic perichondrium that expresses Col1a1(2.3kb)-GFP). The labeling efficiency of *Dlx5-creER* for the inner perichondrium appears to be much higher than 55%. Therefore, although *Dlx5-creER* marks only ~55% of Col1a1-GFP-negative perichondrial cells by our criteria, these cells (*Dlx5*⁺ cells) occupy a defined location within the perichondrium, not randomly distributed across the perichondrium. Again, the initial location of the cells is extremely important in interpreting the lineage-tracing results. The fate of other 45% of perichondrial cells in the outer portion of the perichondrium is beyond the scope of investigation of our current study.

In sum, we do not think that it is a concern that only half of the perichondrial cells can be marked by *Dlx5-creER*. We think it is rather the strength of the study, as *Dlx5-creER* can mark a specific subset of perichondrial cells with specified functions. This notion is supported by the biological significance of these cells, as demonstrated by subsequent analyses.

11. <Reviewer>

****That ~20% of the *Osx*-CE cells label chondrocytes – after only 24 hours of lineage labeling – presents significant challenges in the claim of separation of populations and labeling them as osteogenic perichondrial cells is a bit muddy.**

<Response>

We agree with the reviewer that the interpretation of *Osx*⁺ lineage-tracing results is confounded by the fact that both osteogenic perichondrial cells and late chondrocytes are simultaneously marked by *Osx-creER*. To respect the preceding work by Maes et al. (*Dev Cell*, 2010), which is highly cited in the field, we do not intend to challenge the widely-accepted notion regarding the fate of osteogenic perichondrial cells (osteoblast precursors). We believe that our results from *Dlx5-creER* significantly update our knowledge on

the fate of perichondrial cells, while revising the misconception that might have been derived from previous studies.

12. <Reviewer>

Figure 3 reports highly interesting data on cells moving into the bone marrow space. The authors note the high contribution from both *Osx*-CE and *Fgfr3*-CE, but the data seems to indicate much higher contribution from *Osx*CE. This is not easily consistent with the idea that early bone marrow space is from the cartilage template – it seems that *Fgfr3*-Cre contribution should be higher?

<Response>

We appreciate your positive assessment on our lineage-tracing data presented in Figure 3a. We agree with the reviewer that *Osx-creER*-marked cells interestingly show a higher contribution to cells in the marrow space at E15.5 than those marked by *Fgfr3-creER*.

Our interpretation on this E15.5 data is that *Osx-creER* marks prehypertrophic and hypertrophic chondrocytes that can promptly move into the marrow space, while *Fgfr3-creER* marks proliferating chondrocytes that take some time to move into the marrow space. We added the following sentence for clarification.

The following sentences were added to the Results:

Line 176-181:

*The apparently higher contribution of Osx^{CE} -E12.5 cells to the marrow space than that of $Fgfr3^{CE}$ -E12.5 cells at this stage likely reflects the fact that *Osx-creER* marks prehypertrophic and hypertrophic chondrocytes that can promptly move into the marrow space, while *Fgfr3-creER* marks proliferating chondrocytes that take some time to move into the marrow space. Whether Osx^+ osteogenic perichondrial cells contribute substantially to the marrow space at this stage remains unclear.*

13. <Reviewer>

Further, with labeling at E12.5, much of the cartilage template is unlabeled as it is established prior to E12.5 (as supported in Figure 2). While very interesting data, the interpretation of this data is less clear.

<Response>

We believe that the reviewer refers to the *Osx-creER* lineage-tracing data, in which chondrocytes are marked at E13.5, but not at E15.5 or E18.5, after being pulsed at E12.5. Figure 2 demonstrates that *Osx-creER* marks prehypertrophic and hypertrophic chondrocytes near the center of the cartilage template. These chondrocytes are post-mitotic, do not replicate themselves within the growth plate and eventually take a cell fate of either cell death due to apoptosis or transformation into osteoblasts or marrow stromal cells. The important point from this data is that *Osx-creER* marks transient chondrocytes in the growth plate. This point was clarified in the revised manuscript.

The following underlined sentences were added to the Results:

...highlighting that *Fgfr3-creER* can mark chondroprogenitor cells whereas *Osx-creER* marks prehypertrophic chondrocytes with transient nature.

14. <Reviewer>

In Fig 3b, how did the *Dlx5* lineage come to the center of the trabecular area (perhaps more time points between E12.5 and E15.5 are needed)? This data suggests the Groove of Ranvier cells travel laterally and is inconsistent with cells from this central region being chondrocyte-derived. It would be interesting to note the time points from E12.5 to E18.5 that permits a visualization of the presumed migratory path of these cells.

<Response> (##1)

Thank you for your important comments. We agree with the reviewer's interpretation of our data that perichondrial cells might have moved laterally to occupy the marrow space.

Following the reviewer's suggestions, we performed additional *Dlx5-creER* lineage-tracing experiments to observe additional intermediate time points between E12.5 and E18.5, including E14.5, E16.5 and E17.5 (E13.5, E15.5 and E18.5 were included in the original manuscript).

Figure R1-3 (new Supplementary Figure 2)

Based on these lineage-tracing data, we believe that cells at the Groove of Ranvier do not travel laterally to enter the marrow space. It appears that the cells at the central portion of the perichondrium that move laterally and populate the nascent marrow space.

The following sentences were added to the Results:

Line 194-203:

Additionally, we defined the presumed migratory path of $Dlx5^{CE}$ -E12.5 cells from the perichondrium to the marrow space by analyzing serial time points from E13.5 to E18.5. Consistent with our findings described above, $Dlx5^{CE}$ -E12.5 cells expanded within the $Col1a1$ -GFP^{neg} outer domain of the perichondrium for 2 days (Supplementary Figure 3a,b), and subsequently started to move laterally to the $Col1a1$ -GFP⁺ osteogenic perichondrium after 3 days of chase at E15.5 (Supplementary Figure 3c). After 4 days of chase at E16.5, $Dlx5^{CE}$ -E12.5 cells moved further laterally toward the nascent marrow space and massively expanded therein, and progressively differentiated into $Col1a1(2.3kb)$ -GFP⁺ osteoblasts of the trabecular and cortical compartments (Supplementary Figure 3d,f). These findings demonstrate that $Dlx5^{+}$ perichondrial cells can travel laterally to enter the marrow space.

15. <Reviewer>

It is also intriguing that there are many Osx - and $Fgfr3$ -lineage labeled cells in this region that are not marked with $Col1$. It seems there is an opportunity here, by understanding overlap, how these are related in this area.

<Response>

Thank you for the note. The reviewer is correct in that there are a number of $Dlx5$, Osx and $Fgfr3$ -lineage marked cells in the marrow space that do not express $Col1a1(2.3kb)$ -GFP. We categorized these cells under a broadly blanketed category of “bone marrow stromal cells (BMSCs)”. We believe that some of these cells coincide with CXCL12-abundant reticular (CAR) cells that represent a distinct cell state from bone-forming osteoblasts.

How BMSCs derived from Osx^{+} cells and those $Fgfr3^{+}$ cells are related in the marrow space is an interesting question that is beyond the scope of our current study. We assume that there is a substantial difference between the two BMSCs, as Osx -derived BMSCs include not only those derived from the hypertrophic layer but also those derived from the osteogenic perichondrium, while $Fgfr3$ -derived BMSCs include only those derived from the hypertrophic layer.

16. <Reviewer>

While the authors denote comparable percentages between Dlx - and $Fgfr3$ -Cres, Osx -Cre is not discussed.

<Response>

We have now revised the text to discuss Osx -creER-derived cells in the comparison.

The following underlined sentences were added to the Results:

Line 234:

*...have dual origins in the fetal perichondrium and the cartilage template among *Dlx5-creER*⁺ cells and *Fgfr3-creER*⁺ cells, respectively, while *Osx-creER*⁺ cells overlap with these cell types.*

17. <Reviewer>

With the amount of overlap in other areas noted and *Dlx* only labeling 50% of perichondrial cells, the conclusion that trabecular osteoblasts derive 50/50 from perichondrial and chondrocyte sources seems over-interpreted (and potentially internally inconsistent).

<Response>

Thank you for the comment. We have clarified the sentence in the revised manuscript and dampened the conclusion.

The following underlined words were added to the Results:

Line 217-219:

*...indicating that trabecular bone osteoblasts at this stage might be equally derived from the two cells-of-origins of *Dlx5*⁺ perichondrial cells and *Fgfr3*⁺ chondrocytes.*

18. <Reviewer>

Further, the absolute number of cells contributed by each Cre lineage does not allow a comparison, particularly given the differential efficiencies and/or limitation of the Cres and the inability to assess a total population (part of Fig 3c) - this does not lend to a meaningful use of total numbers to reflect a direct comparison.

<Response>

Thank you for the note. We agree with the reviewer that the contribution of a particular cell type to another cell type can be either overestimated or underestimated due to different efficiencies of *creER*s.

However, we would like to emphasize that *Dlx5-creER* and *Fgfr3-creER* mark a similar fraction of the parental cell populations, that is, ~55% of *Col1a1-GFP*^{negative} perichondrial cells and ~40% of *SOX9*⁺ chondrocytes of the cartilage template, respectively (**Figure 2c**). In addition, these *creER* lines do not randomly mark their parental cell populations, but rather mark a specific group of cells at a specific location, that is, the specific layer of the perichondrium and proliferating chondrocytes, respectively (**Figure 2b**). Therefore, we believe a direct comparison between these two cell types is valid owing to similar efficiencies of *creER* lines.

In addition, we mitigated the potential bias by enumerating the lineage contribution in percentages when possible (**Figure 3c**), except BMSCs. We were unable to enumerate the lineage contribution in percentages for BMSCs, because we could not properly define the parental cell population because of the anatomical complexity of the marrow space.

Following the reviewer's comments, we have now added the following sentence in the revised manuscript to acknowledge the potential limitation.

The following underlined sentences were added to the Results:

Line 225-226:

...indicating that *Dlx5*⁺ early perichondrial cells may preferentially contribute to marrow stromal cells despite a potential bias in cell counting.

19. <Reviewer>

Figure 4g is a strikingly interesting result and deserves significantly more attention (for which room could be made by moving some of the earlier panels to supplemental).

<Response>

Thank you for your positive assessment on our functional study presented in Figure 4. We have added new data and rearranged this figure following the reviewer's comments, as detailed in Point #20.

20. <Reviewer>

Here, the phenotype in *Fgfr3-Ptc1* mutants (and *Dlx-Ptc1*) is worthy of further investigation and incomplete as presented. It is difficult to understand how bm space can be reduced and cartilage length not increased with total lengths not being different. For each N, it may be more appropriate to normalize bone marrow length/total length to ensure this is not a rigor issue with Ns being too low to cause significantly lower bone length but having barely significant changes in bm length. It seems there is a missing piece of information here that might require more Ns or normalized measurements.

<Response> (##1)

We appreciate your important insights. We have now included additional N's in each group (*Dlx5*-Cont: *n*=8 [originally *n*=5]; *Dlx5*-cKO: *n*=11 [originally *n*=5]; *Fgfr3*-Cont: *n*=10 [originally *n*=5]; *Fgfr3*-cKO: *n*=8 [originally *n*=5]). We further normalized the bone marrow length and the distal cartilage length by total length, as suggested by the reviewer.

Figure R1-4 (revised Figure 4g, the second column from the right)

21. <Reviewer>

For both mutants, assessing the changes in trabecular length, length of differentiating cartilage columns and other measures that further define what the Hh GOF is causing is critical to understanding the phenotype; this phenotype is under-assessed.

<Response>

Thank you again for your important suggestions. We have now added three parameters in the revised manuscript – (1) *Trabecular area per marrow area*, (2) *Number of chondrocytes per column* and (3) *Total number of columns in growth plate*, as suggested by the reviewer. We have also appended the interpretation of these results in the revised text.

Figure R1-5 (revised Figure 4g, the first column from the right)

The following sentences were added to the Results:

Line 278-282:

Further, the Col1a1(2.3kb)-GFP⁺ trabecular area per the marrow area was significantly reduced in Fgfr3-Ptch1 cKO, associated with an increase in the number of chondrocytes in each column while the total number of chondrocyte columns was unchanged (Fig.4g, first column from the right), further suggesting a delay in the chondrocyte-to-osteoblast transition.

22. <Reviewer>

Again, the interpretation that Dlx cells regulate bone marrow formation without the chondrocytes is a very liberal 'suggestion' from this panel of information. Much more could be learned here without significant extra effort on the sections from these animals.

<Response>

Thank you again for your important suggestions. Based on the new quantitative data of the Hedgehog gain-of-function study described above, we have now added the following sentences in the revised results.

The following underlined sentences were added to the Results:

Line 282-285:

Therefore, Dlx⁵⁺ early perichondrial cells directly regulate the formation of the marrow space, whereas Fgfr3⁺ fetal chondrocytes regulate bone formation within the marrow space through chondrocyte-to-osteoblast transition, at least in part mediated through Hedgehog signaling

23. <Reviewer>

Fig 5 shows a relatively expected difference of contribution to bm reticular cells. However, there are two important things this data reveals that the authors do not comment on and is critical for the overall interpretation of their data. The first is the apparent relatively low contribution to osteocytes. This is not quantified (but could be). The data shown shows surprisingly little contribution from either Cre – not consistent with having identified the two sources of osteoblasts and reticular cells.

<Response>

Thank you for your comments. The quantification of lineage-marked tdTomato⁺ osteoblasts/osteocytes in each model was included in the original manuscript in **Figure 5**. We apologize for the lack of clarification. These data demonstrate that both Dlx5^{CE}-E12.5 and Fgfr3^{CE}-E12.5 cells contribute to a substantial number of osteocytes, albeit in different locations (diaphysis and metaphysis, respectively).

Figure R1-6 (revised Figure 5, Supplementary Figure 3)

The following underlined word was added to the Results:

Line 309:

Dlx5^{CE}-E12.5 perichondrial cells predominantly contributed to BMSCs and osteoblasts and osteocytes in the distal area (3-7mm from the growth plate), whereas Fgfr3^{CE}-E12.5 cells contributed to these cells in the proximal area (0-3mm from the growth plate, Fig.5c).

The following sentences were added to the Results:

Line 320-322:

Similarly, not all the osteocytes were derived from either Dlx5^{CE}-E12.5 or Fgfr3^{CE}-E12.5 cells, suggesting that other perichondrial cells or chondrocytes may contribute to the remaining fraction.

24. <Reviewer>

This is in contrast to the report using *Hoxa11Cre* which shows complete labeling of osteoblasts and osteocytes where the perichondrium is labeled (Pineault, 2019; Song, 2020). Perhaps this is due to only 50% of the perichondrium being labeled with this Cre?

<Response>

Thank you again for raising this point out for discussion. The contrast between our results and the *Hoxa11* study is interesting. As discussed above, in the zeugopod, *Hoxa11-creER* appears to mark both perichondrial cells and chondrocytes within the cartilage template, which may partly explain their efficient contribution to osteocytes. We would like to emphasize that our study weighs more on subset specificity, rather than efficiency to label the entirety of the lineages that is the focus of the *Hoxa11* study.

Additionally, our quantification on osteocytes demonstrate that *Dlx5*⁺ and *Fgfr3*⁺ cells contribute robustly to osteocytes, particularly those in the diaphysis and the metaphysis, respectively. However, it is possible that remaining 45% of *Dlx5*-negative perichondrial cells also contribute to osteocytes. We have clarified this point in the revised manuscript.

The following sentences were added to the Results:

Line 320-322:

*Similarly, not all the osteocytes were derived from either *Dlx5*^{CE}-E12.5 or *Fgfr3*^{CE}-E12.5 cells, suggesting that other perichondrial cells or chondrocytes may contribute to the remaining fraction.*

25. <Reviewer>

Related, the endosteum, a source of osteoblasts and the majority of cortical bone growth as shown by several recent publications, remain strikingly unlabeled, perhaps accounting for this 'lost' pool of progenitors.

<Response>

Thank you very much for your note on this very important finding. We have now added this point to the revised manuscript.

The following underlined clauses were added to the Results:

Line 304-306:

*Interestingly, both *Dlx5*^{CE}-E12.5 cells and *Fgfr3*^{CE}-E12.5 cells had disappeared from the perichondrium (groove of Ranvier) by this stage, and appeared to make only minimal contribution to the endosteum.*

26. <Reviewer>

The authors' report that 19% of the reticular cells are from *Dlx*-Cre lineage and 26% from the *Fgfr*-Cre lineage. This is less than 50% of total? Drawing conclusions on discovering the two sources of reticular

cells without accounting for missing information (the other 60% of the Cxcl12+ cells) should be addressed or clarified.

<Response>

Thank you again for pointing out this important point for discussion. We have now addressed this point in the revised manuscript.

The following sentences were added to the Results:

Line 317-319:

The remaining fraction of Cxcl12-GFP^{high} stromal cells might be derived from other perichondrial cells or chondrocytes at E12.5 that were not marked either by Dlx5-creER or Fgfr3-creER.

27. <Reviewer>

While it may have been published that Col2a1-Cre is a chondrocyte subset, the marking of perichondrial cells suggest it is a more general marker of cells that express collagen 2 – connective tissue. This author agrees that Col2a1 is not a particularly informative lineage marker, but this data and discussion can probably be skipped as it does not add value.

<Response>

The reviewer's point is well-taken. However, we believe that the *Col2a1-creER* lineage-tracing data is extremely valuable to justify the use of the two lines (*Dlx5-creER* and *Fgfr3-creER*) to deconvolute the lineage contribution of its subsets. We therefore opted to keep these data in the Supplemental Figure.

28. <Reviewer>

The bulk-seq data reported in Figure 6 is interesting. The gene expression shown in Fig 6c is also interesting. As changes are small (not surprising), it seems relevant to show these genes in some context of what biggest changes are between the two populations in addition to the ones the authors picked out to highlight. Some presentation of the more complete and unbiased assessment of the data needs to be shown and discussed in the context of the gene expression changes they chose to highlight.

<Response>

Thank you very much for your positive assessment on our comparative RNA-seq analysis. We have now reanalyzed the bulk RNA-seq data and included more complete and unbiased sets of data in the revised manuscript, including the volcano plot and Gene Ontology enrichment analysis, as requested by the reviewer.

One of the most important technical hurdles of our current comparative bulk RNA-seq analysis is that the two datasets (*Dlx5*^{E12.5}-Cxcl12-GFP⁺ and *Fgfr3*^{E12.5}-Cxcl12-GFP⁺) were generated by two different platforms (NovaSeq 6000 / PE150 and HiSeq 4000 / SE50, respectively). Therefore, we have also performed additional "clean-up" steps to streamline the data analysis. We have also included these details in the revised manuscript.

Figure R1-7 (revised Figure 6)

The following sentences were added to the Results:

Line 358-360:

...demonstrating the unique molecular characteristics of these two cell types. Accordingly, a large number of genes were differentially expressed between the two cell types, with ~1,400 and ~1,800 genes upregulated in Dlx5^{CE}-E12.5 and Fgfr3^{CE}-E12.5 cells, respectively (Fig.6c).

Line 366-374:

Gene Ontology (GO) enrichment analysis of differentially expressed genes revealed that significant enrichment of biologically relevant GO terms in Dlx5^{CE}-E12.5 cells, including negative regulation of cell migration (GO:0030336) and fat cell differentiation (GO:0045444) and ossification (GO:0001503) (Fig.6e), revealing the fundamental molecular identity of Dlx5⁺ perichondrium-derived BMSCs. Conversely, many cell division-related GO terms [mitotic cell cycle phase transition (GO:0044772), mitotic nuclear division (GO:0140014) and ossification (GO:0001503)] were enriched in Fgfr3^{CE}-E12.5 cells (Fig.6e), revealing the actively cycling nature of Fgfr3⁺ fetal cartilage-derived BMSCs.

The following underlined sentences were added to the Methods:

...and submitted for NextGen sequencing (Illumina NovaSeq 6000 150 base for pair-end reads for Dlx5^{E12.5} samples [n=3 biological replicates], HiSeq 4000, 50 base for single-end reads for Fgfr3^{E12.5} samples [n=2 biological replicates]).

29. <Reviewer>

The CFU-F assay goes back to looking at only Fgfr3-Cre and Dlx-Cre lineage bm cells – how many of these are Cxcl12-positive in each set? Was this stated? The relationship to bigger story seems parenthetical. Certainly a paragraph following this data to state that the theory presented in the previous paragraph is supported could be removed.

<Response>

Following the reviewer's requests, we have now included the percentage of Cxcl12-GFP⁺ cells among Dlx5-creER and Fgfr3-creER lineage-marked cells in the revised manuscript (Cxcl12-GFP⁺ cells: 66.2±12.6% of Dlx5^{CE}-E12.5 cells, 17.3±4.1% of Fgfr3^{CE}-E12.5 cells at P21). The percentage of Cxcl12-GFP⁺ cells in each subset does not appear to correlate with their clonogenic activities.

We did not perform Osx-creER marked cells for CFU-F assays, because Osx-creER lineage cells contribute to a very small percentage of BMSCs at this stage, accounting for only 1% of Cxcl12-GFP⁺ stromal cells (in contrast to 19% and 26% by Dlx5-creER and Fgfr3-creER lineage cells, respectively).

We opted to keep the paragraph that the reviewer mentioned in the revised manuscript, as it pertains to the major conclusion of this study.

The following underlined sentences were added to the Results:

Line 380-382:

Flow cytometry analysis revealed that Cxcl12-GFP^{high} cells represented 66.2±12.6% and 17.3±4.1% of Dlx5^{CE}-E12.5 and Fgfr3^{CE}-E12.5 cells, respectively, at P21 (Fig.5d), which does not appear to correlate with their clonogenic activities. These results are consistent with our recent finding that the adipogenic subset of CXCL12^{high} stromal cells (Cxcl12-creER⁺ stromal cells³⁶, akin to Adipo-CAR cells³⁷) have little colony-forming activities.

30. <Reviewer>

Figure 7 should be removed in its entirety. The focus of this study on Pth is peripheral and the later time points of investigation, well past when the two Cres have separated almost entirely spatially, make for a messy, unrelated, but perhaps beginning of an interesting story discussing what genetic perturbation of half of the perichondrium and most of the cartilage causes. It is premature as presented currently.

<Response>

Thank you for your critical note. However, we believe that the iPTH experiment in Figure 7 is essential for supporting our conclusion that Fgfr3-creER-lineage BMSCs are more prone for osteogenesis compared to Dlx5-creER-lineage BMSCs. Importantly, this experiment provides biological validation for our computational prediction from our comparative bulk RNA-seq analysis, regarding adipocyte-biased state of Dlx5-creER-lineage cells. We designed PTH experiments to bring out the adipogenic and osteogenic nature of BMSCs derived from the two cellular sources.

In light of the positive comments from the other reviewers, we opted to keep this figure in the revised manuscript.

31. <Reviewer>

Discussion:

The work shared here does not provide a cell of origin for the periosteum as stated. The DlxCre does mark some of the perichondrium, at a stage where perichondrial cells are already present. It does not speak to their origin; this may have been misstated.

<Response>

Following the reviewer's request, we have now removed the strike-throughed clause in the discussion, "...an important cell-of-origin for ~~not only the periosteum but also the distal bone marrow stroma...~~" (Line 424).

Response to Reviewer #2

Thank you very much for your constructive comments and critiques.

1. <Reviewer>

In this manuscript, Matsushita et al. examine Dlx5 versus Fgfr3-lineage cells using a creERT based lineage tracing approach as a means to examine the eventual fates and roles of cells in the early developmental perichondrium versus the developmental cartilage template. Taken together, this manuscript addresses a novel area of the literature in examining the function and fates of the early perichondrium, and identifies several important distinctions between the derivatives of the perichondrial cells versus other skeletal populations. Perhaps chief among these and the most interesting finding reported here is that this data suggests distinct developmental origins for the recently described osteogenic and adipogenic subsets of Cxcl12 abundant reticular cells (CAR cells). This is particularly important as one of the central questions regarding these two cell types is whether they represent the recent bifurcation of a single progenitor present in adults that gives rise to both cell types or whether these are two wholly distinct cell lineages with distinct origins that happen to share some markers in common. The data presented here argues for perhaps the more surprising and interesting latter possibility. Additionally, while there are areas that require qualification or clarification as indicated below, the methods used are robust and appropriate. Taken together, there is overall enthusiasm for the importance and novelty of this study.

<Response>

Thank you very much for your positive assessment on our work. We have incorporated the reviewer's great summary sentences in the revised discussion.

The following sentences were added to the Discussion:

Line 443-447:

One of the central questions regarding the two CAR cell types is whether these cells represent the recent bifurcation of a single progenitor present in adults that gives rise to both cell types or whether these are two wholly distinct cell lineages with distinct origins that happen to share some markers in common. The data presented here argues for the latter possibility...

2. <Reviewer>

Major point

1. It is an interesting, and likely very important, finding that dlx5 and fgfr3 account for distinct groups of CAR cells that appear by expression profile to correlate with adipo-CAR and osteo-CAR populations. However, this finding would be greatly strengthened if it was demonstrated to correlate with an eventual maturation of the dlx5-lineage putative adipo CAR cells into marrow adipocytes, and conversely maturation of the FGFR3-lineage putative osteo-CAR cells into marrow/trabecular osteoblasts with extended chase periods.

<Response>

Thank you very much for your comments. Following the reviewer's suggestion, we have now performed additional immunohistochemical staining at 3M to demonstrate eventual maturation of these cells. We performed perilipin (PLIN) and alkaline phosphatase (ALPL) staining to label bone marrow adipocytes and osteoblasts, respectively.

Figure R2-1 (new Supplementary Figure 3)

The following underlined sentences were added to the Results:

Line 325-328:

At three months of age, *Dlx5*^{CE}-E12.5 cells were present in the diaphyseal marrow stromal compartment and contributed to Perilipin⁺ marrow adipocytes, while *Fgfr3*^{CE}-E12.5 cells were present in the metaphyseal marrow stromal compartment and contributed to Alkaline phosphatase⁺ osteoblast-like cells on the bone surface.

3. <Reviewer>

Can these groupings be observed to correlate with the osteolectin-based classification of CAR subsets proposed in Shen et al. Nature 2021? In general, comparison to the Shen et al. paper as well as deeper comparison to the Baccin et al Nat Cell Biol2020 paper is warranted. The Shen et al. report should be cited.

<Response> (##2)

We thank the reviewer for these excellent points for discussion. We agree with the reviewer that it is important to cite the Shen report in place our work in full context. We have now added the following sentences in the revised manuscript, to discuss how our findings could correlate with the Clec11a/Osteolectin-based classification of CAR cell subsets defined by the osteolectin study (Shen et al. 2021) and the spatial transcriptomic study (Baccin et al. 2020). We cited the Shen study along with the discussion.

The following sentences were added to the Discussion:

Line 450-460:

Recent investigation demonstrates that leptin receptor (*LepR*)-expressing marrow stromal cells are composed of two distinct peri-arteriolar and peri-sinusoidal subsets with different functionality. The peri-arteriolar *LepR*⁺ cells express ostelectin (encoded by *Clec11a*), which are rapidly-dividing and short-lived and poised to osteogenesis in a mechanosensitive manner⁴⁰. This osteolectin⁺ subset of *LepR*⁺ cells may coincide with Osteo-CAR cells localized to peri-arteriolar surfaces and also act as professional cytokine-secreting cells³⁷. Importantly, these studies also indicate that bone marrow stromal cells poised for osteogenesis and adipogenesis represent two distinct cell lineages, as these *LepR*⁺osteolectin⁺ cells are not fated to adipocytes⁴⁰ and Osteo-CAR cells are not computationally predicted to give rise to Adipo-CAR cells³⁷. It remains to be determined whether *Fgfr3*⁺ fetal chondrocyte-derived CAR cells are similarly rapidly-dividing, short-lived and preferentially localized to peri-arteriolar surfaces.

Additionally, we also investigated the expression level of *Clec11a* mRNA based on our comparative bulk RNA-seq experiment shown in Figure 6. There was no statistical significance between *Dlx5*^{CE}-E12.5 and *Fgfr3*^{CE}-E12.5 *Cxcl12*⁺ stromal cells.

Figure R2-2

4. <Reviewer>

2. An important limitation of using this creERT-based lineage tracing approach to assess cellular translocation, especially over the fairly fast progression of early murine skeletal development, is that it is difficult to exclude that there is additional new recombination occurring up to several days after the initial tamoxifen injection in the populations apparently translocating. This is especially a concern given the demonstrated long half-life of tamoxifen. It doesn't appear that any of the data presented strongly excludes this possibility. It is appreciated that it is difficult to formally exclude this possibility, so it would be sufficient to acknowledge this limitation prominently.

<Response>

Thank you very much for your comments. We acknowledge the potential limitation of our tamoxifen-inducible *creER*-based lineage-tracing approaches is that additional new recombination may occur for several days after tamoxifen injection due to a relatively long half-life of tamoxifen (generally 24-48 hours), while cells are actively translocating to the nascent marrow space. We have added the following sentence in the revised manuscript to prominently acknowledge this limitation.

The following sentences were added to the Discussion:

Line 473-477:

A potential limitation of our tamoxifen-inducible creER-based lineage-tracing approaches is that additional new recombination may occur for several days after tamoxifen injection, while cells are actively translocating to the nascent marrow space. It is therefore technically challenging to formally distinguish the cell fates of a cell population and its immediate descendants in a fast-evolving process of early bone development. Despite these limitations, we believe that [our findings indicate that...]

5. <Reviewer>

Minor points and discussion:

1. In the Fig 1 single cell seq study, how is it determined which clusters represent chondrocytes and which clusters represent perichondrial cells? As the authors are aware, the expression of classically chondrocyte associated transcripts in perichondrial cells make this distinction not straightforward on the basis of transcriptional data alone.

<Response> (##1)

Thank you for asking this important question. To further strengthen the justification on the classification of chondrocytes and perichondrial cells in Figure 1, we additionally performed transgenic reporter assays, RNAScope assays and immunohistochemistry of representative genes to correlate our transcriptional data to in situ spatial expression patterns. For this purpose, we validated expression of five marker genes, including SOX9 (IHC), *Fgfr3* (GFP reporter), *Sp7/Osx* (mCherry reporter), *Col2a1* and *Dlx5* (RNAScope).

Figure R2-2 (part of new Figure 1)

We found that in situ expression patterns of these marker genes correlate fairly well with our designation of chondrocytes and perichondrial cells in the single-cell RNA-seq dataset in Figure 1.

As expected from the single-cell RNA-seq expression data, SOX9 protein expression was essentially confined to fetal chondrocytes within cartilage templates. Further, Fgfr3-GFP reporter activities were confined to a subset of SOX9⁺ chondrocytes that are relatively more mature and differentiated located toward the center of the cartilage template. In contrast, Sp7-mCherry activities were evident not only in the perichondrium but also in the cartilage template representing an even more mature and differentiated subset of SOX9⁺Fgfr3⁺ chondrocytes in the center. RNAScope analyses revealed that, while *Col2a1* mRNA was almost exclusively expressed within the cartilage template, *Dlx5* mRNA was predominantly found in the perichondrium. These in situ expression data have been included in the revised manuscript.

The following sentences were added to the Results:

Line 105-116:

We further examined in vivo expression patterns of potential cell type-specific genes that we identified in our scRNA-seq analyses using independent approaches, including Fgfr3-GFP; Sp7-mCherry double transgenic reporters, RNAScope assays and SOX9 immunostaining. Importantly, SOX9 proteins and Fgfr3-GFP (MMRRC:031901) activities were essentially confined to fetal chondrocytes within cartilage templates, with Fgfr3-GFP⁺ cells representing a more differentiated subset of SOX9⁺ cells located toward the center of the cartilage. Sp7-mCherry activities were evident not only in the perichondrium but also in the cartilage template representing an even more differentiated subset of SOX9⁺Fgfr3⁺ chondrocytes. Additionally, while Col2a1 mRNA was almost exclusively expressed within the cartilage template, Dlx5 mRNA was predominantly expressed in the perichondrium. Therefore, these in vivo gene expression patterns are concordant with our cluster designation as chondrocytes and perichondrial cells in the scRNA-seq dataset.

6. <Reviewer>

2. Previously the authors in the Ono et al. Nat Cell Biol 2014 had used studies of a Sox9-creERT to generate data supporting the conclusion that long-lived periosteal/perichondrial progenitors/stem cells express Sox9. How do the authors interpret the apparent exclusion of Sox9 from the Dlx5 population in light of this prior data? Is Sox9 still a marker of skeletal stem cells in this context? It is appreciated that the setting being studied here is different than that in earlier studies and that no formal claims are made here about the Dlx5-lineage cells including a stem cell, nevertheless this finding was unexpected.

<Response>

Thank you for these important comments. Interestingly, we notice a significant discrepancy between Sox9-creER activities and SOX9 protein expression in the fetal cartilage. As we reported previously, Sox9-creER marks both perichondrial cells and fetal chondrocytes within the cartilage template at the same stage. However, SOX9 proteins are expressed in a more restricted manner only by chondrocytes within the cartilage template.

Figure R2-3

The reason underlying this apparent discrepancy is unknown, although it is possible that post-transcriptional and post-translational mechanisms are at work to inhibit SOX9 protein expression in the perichondrium. We therefore speculate that Sox9-creER-marked cells may encompass both Dlx5-creER and Fgfr3-creER-marked cells in the fetal cartilage, in the same way as Col2a1-creER-marked cells do, which we showed in **Supplementary Figure 5**.

Whether Sox9 can serve as a marker for skeletal stem cells awaits further investigation. We would like to point out that, in our previous work, we marked Sox9-creER⁺ cells at a later stage, at postnatal day 3, when Sox9-creER activities become much more restrictive to the growth plate.

7. <Reviewer>

3. Is the low numbers of DLX5 lineage cells that become col1 expressing osteoblasts just a factor of the timepoint chosen or is this more reflective of an intrinsically low rate of these cells contributing to the osteoblast pool?

<Response>

Thank you for your question. As we demonstrated in Figure 3b,c, Dlx5-creER-lineage cells could initially contribute very well to Col1a1(2.3kb)-GFP⁺ osteoblasts in the cortical bone at E18.5 (~70%). However, this contribution appears to be exhausted at later time points, as these cells did not contribute very well to cortical osteoblasts in the metaphysis at P21. We think this is due to the transient nature of Dlx5⁺ lineage cells that cannot maintain their pool of osteoprogenitor cells for a long period in the metaphyseal marrow space, which are gradually replaced by other cellular sources, such as Fgfr3⁺ lineage cells in postnatal stages.

We clarified this point in the revised manuscript. Thank you for this discussion.

The following sentences were added to the Discussion:

Line 479-486:

Interestingly, the contribution of $Dlx5^+$ cell derivatives to cortical bone and marrow stromal compartments of the metaphyseal marrow space appears to be exhausted at later postnatal stages. We assume that this is due to the cell fate of $Dlx5^+$ perichondrial cells that are primarily destined to become long-lived adipocyte-biased marrow stromal cells, which do not readily convert into short-lived osteoblast-biased marrow stromal cells. We therefore postulate that the $Dlx5^+$ cell-derived osteoprogenitor pool in the marrow space is gradually replaced by those derived from other cellular sources such as $Fgfr3^+$ cells in the metaphyseal marrow space in postnatal stages.

8. <Reviewer>

4. The wording on line 152 implies that the osx -lineage cells in the perichondrium are moving into the marrow space. However, it would appear from the data to be just as likely, or perhaps more likely, that the osx -lineage cells populating the marrow are derived from osx -expressing cells within the cartilage template, not the perichondrium. It is suggested that this wording be adjusted to avoid indicating translocation or at least to indicate the uncertainty regarding this.

<Response>

Following the reviewer's comments, we have now replaced the word "*translocated into*" with "*populated*" when referring to Osx^+ and $Fgfr3^+$ cells within the cartilage template in the revised manuscript (Line 174).

The reviewer is correct that it is more likely that Osx -creER lineage cells that are populating the marrow space are those derived from the cartilage template, including prehypertrophic and hypertrophic chondrocytes. We have emphasized the uncertainty regarding the identity of Osx^+ cells rapidly translocating to the marrow space.

The following sentences were added to the Results:

Line 180-181:

Whether Osx^+ osteogenic perichondrial cells contribute substantially to the marrow space at this stage remains unclear.

9. <Reviewer>

5. Caution is advised regarding the use of the term "undifferentiated" with reference to Fig 2. The $Col1a1$ -negative $DLX5$ -lineage cells may be partially differentiated or differentiated to a non-osteoblast fate. Similarly, it is recommended that a more plainly descriptive term (e.g, $col1a1$ -negative) be used instead of undifferentiated throughout.

<Response>

Thank you very much for this suggestion. We have replaced the term "*undifferentiated*" with "*Col1a1-GFP-negative*" throughout the revised manuscript.

10. <Reviewer>

6. When the authors use the term "cortical osteoblasts" such as in line 170, is that used as a synonym for "periosteal osteoblasts" or "periosteal and endosteal osteoblasts"? It would be helpful to be explicit about which sides of the cortex are being labeled.

<Response>

It is extremely difficult (almost impossible) for us to distinguish which side of the cortex are being labeled at E18.5, as the cortical bone is extremely thin at this stage. Even at P21, we could not explicitly determine which sides of the cortical bone is being labeled.

We have clarified these cells as "*cortical bone osteoblasts (periosteal and endosteal osteoblasts)*" in the revised manuscript, following the suggestion of the reviewer. Thank you for this suggestion.

11. <Reviewer>

7. Optionally, it may be easier to follow the discussion starting in line 207 if annotations are used to refer to the cluster (e.g., Dlx5 and Fgfr3 clusters) names rather than the numbers assigned.

<Response>

We have revised the cluster annotations, as suggested by the reviewer.

12. <Reviewer>

8. In the Ono et al. Nat Cell Biol 2014 report, the *Osx*-creERT tracing did label CXCL12 abundant reticular cells. Is lack of similar labeling noted in line 250 just due to differences in timing between the present study and the author's past work? If so, that might be a helpful point to cover in the discussion.

<Response>

Thank you for raising this important point for discussion. In our previous work (Ono et al. 2014), we demonstrated that cells marked by *Osx-creER* at E13.5 (*Osx*^{CE}-E13.5 cells) disappear almost completely from the marrow space at P21, whereas *Osx*^{CE}-P3 cells initially contribute to CXCL12-GFP⁺ cells (~4 weeks) then also gradually disappear from this fraction.

Therefore, our current findings are consistent with our previous findings, in that *Osx*⁺ cells contribute only transiently to CXCL12-abundant reticular cells, and eventually disappear from this fraction. We have added the following sentence in the discussion to clarify this point.

The following sentences were added to the Discussion:

Line 486-489:

Additionally, Osx is also expressed by an osteoblast-biased subset of CAR cells within the postnatal marrow space; however, these Osx⁺CXCL12⁺ marrow stromal cells are also transient and gradually disappear from this fraction²², presumably replaced by other cellular sources such as Fgfr3⁺ cells.

13. <Reviewer>

9. It would be helpful for the authors to compare the labeling seen with FGFR3 versus their prior work with PTHrP-creERT. Are differences in the marrow stromal populations labeled due to technical differences in these two cre lines or biologic differences in the labeled populations? Underlying this question is a desire to understand the parallels between the cellular composition and eventual fates of the initial cartilage template versus the later growth plate resident stem cells. Given the complexity of the literature on skeletal cell types, in general expansion of the discussion to more strongly tie the current results to past studies would strengthen the manuscript.

<Response> (##2)

Thank you very much for these great suggestions. We agree with the reviewer that understanding similarities and differences in the marrow stromal cell fates of Fgfr3⁺ chondrocytes in the fetal cartilage (current work) and PTHrP⁺ chondrocytes of the resting zone of the postnatal growth plate (previous work) is important. We strongly believe that there are substantial biological differences between these two types of “stem cells” (cartilage template versus later growth plate), which make their eventual cell fates fundamentally different.

First, our current work demonstrates that Fgfr3⁺ fetal chondrocytes can exuberantly translocate into the marrow space and continue to feed into osteoprogenitor cells in the postnatal stage, whereas PTHrP⁺ resting chondrocytes only moderately contribute to marrow stromal cells and osteoblasts in the marrow space. Therefore, it is likely that fetal chondrocytes exhibit far superior capability to generate multitudes of skeletal cell types, unlike postnatal chondrocytes. The mechanism that confers fetal chondrocytes the capability to transform into osteoprogenitor cells is unclear at this stage, and represents an important agenda for future investigation.

We have added the following sentences in the revised discussion.

The following sentences were added to the Discussion:

Line 461-467:

Of note, Fgfr3⁺ fetal chondrocytes of the cartilage template appear to generate osteoblasts and stromal cells much more efficiently than PTHrP⁺ resting chondrocytes of the postnatal growth plate do⁴¹. We postulate that substantial biological differences lie between these two types of stem cells – stem cells in the cartilage template and those in the postnatal growth plate – which make their eventual cell fates fundamentally different. The mechanism conferring Fgfr3⁺ fetal chondrocytes the far superior capability to transform into osteoprogenitor cells is unclear at this stage, representing an important area for future investigation.

14. <Reviewer>

10. The description of the scRNA-seq analysis is generally good, however it would be helpful to clarify whether cell cycle state was regressed out and how cell cycle state parsed out among clusters. It would also

be helpful to visualize the number of UMIs or unique genes per cell to see if sampling issues are driving any of the clustering.

<Response>

Thank you very much for your suggestions. We have clarified in the revised manuscript that cell cycle was regressed out in our dataset (*Line 91*). We have also included other relevant data, such as distribution of cells in different cell cycles, # of UMI and unique genes per cell, in the revised manuscript.

The cell cycle states based on S.Score and G2M.Score suggest that cells in different phases of the cell cycle are equally parsed out among clusters, indicating that the cell cycle was properly regressed in this dataset.

Figure R2-4 (new Supplementary Figure 1)

15. <Reviewer>

11. The statement "Our findings therefore indicate that there is a self-renewing cell population within the fetal perichondrium" may not be justified by the data. No formal self-renewal studies were performed, and no data is provided to show a specific population of Dlx5-lineage cells that retains their defining characteristics over time, such as through serial transplantation. Additionally, the timing of chase periods analyzed here is fairly short. This statement should be softened.

<Response>

We appreciate your critical assessment. We have now replaced "self-renewing" with "long-living", and softened the statement, as suggested by the reviewer (*Line 478*).

Response to Reviewer #3

Thank you very much for your constructive comments and critiques.

1. <Reviewer>

In this paper, Matsushita et al perform lineage tracing studies on bone formation in mice using two novel marker genes identified through an initial exploration of an scRNA-seq dataset. Through complex but beautifully designed experiments the authors make use of conditional reporters and knockouts to show the differences in contributions from Dlx5+ and Fgfr3+ cells in formation and development of bone marrow. Further, again building on a theory developed from the scRNA-seq analysis, the authors investigate the role of hedgehog signaling between the involved cell types.

Ideally more biological replicates would have been used, but there are no substantial issues with this manuscript. At times the logic for the results from the lineage tracing experiments can be hard to follow due to the many factors involved (multiple conditional reporters, multiple locations, etc). This is inherent in the complexity of the system studied, but perhaps simple illustrations could be used to summarize the conclusions for each figure?

<Response> (##3)

Thank you very much for your positive assessment on our work. To address potential concerns on biological replicates, we have increased N's when possible (**Figure 4**) to increase the confidence on the findings.

Additionally, we added simple illustrations to summarize the findings in each figure to facilitate the readers' understanding (**Figure 3d, Supplementary Figure 3c**).

2. <Reviewer>

There are a few minor presentation issues:

Panels showing gene expression have no units: Fig. 1e, Fig. 4e,f. Are these counts of UMIs?

<Response>

Thank you for your question. The unit of the scale represents relative expression levels. We have added this information in the revised manuscript.

3. <Reviewer>

In the text, the sub-panels of Fig.3c are referred to as 'leftmost panel', 'left center panel', 'center panel', 'right center panel', and 'rightmost panel'. It would be more clear to refer to these as 'leftmost', 'second from the left', 'third from the left', 'fourth from the left', and 'rightmost panel'.

<Response>

Thank you very much for your suggestions. We have revised the text. As we re-aligned the figure, we named 'upper left', 'upper center', 'upper right', 'lower left' and 'lower right'.

4. <Reviewer>

The text refers to 'Fig.4 red dotted area', but it is not clear what this refers to in the figure.

<Response>

We have revised the word to "red dotted *box*" in the revised manuscript for better clarification.

5. <Reviewer>

Also in Fig. 4a, what unit is 'interaction strength'?

<Response>

Interaction strength of each cluster was computed by intercellular communication probability, after considering of not only ligand-receptor interactions, but also multimeric receptors, soluble agonists and antagonists, co-stimulatory, and co-inhibitory membrane-bound receptors. Therefore, the interaction strength shown on the x- and y-axis in Fig 4a is unitless. With that in mind, potential readers of this paper are advised to assess these relative values by considering how one cluster's incoming and outgoing interaction strength (e.g. C01) is higher or lower, comparatively, than those of another cluster (e.g. C08), when evaluating the probability of ligand-receptor activities involved in overall and Hh pathways.

6. <Reviewer>

Is there a reference for the fact that deletion of Ptch1 leads to constitutive activation of Hedgehog signaling?

<Response> (##2)

Thank you for your question. We have now added the following references in the revised main text.

- Hahn, Bale, 1996 Cell: "Mutations of the human homolog of Drosophila patched in the nevoid basal cell carcinoma syndrome" (Ref.31)
- Johnson, Scott, 1996 Science: "Human homolog of patched, a candidate gene for the basal cell nevus syndrome" (Ref.32)
- Lum, Beachy, 2004 Science: "The Hedgehog response network: sensors, switches, and routers" (Ref.33)

7. <Reviewer>

In the methods section it says cell numbers were quantified by a 'Countless II' instrument. The correct name is 'Countess II'.

<Response>

Thank you very much. We have corrected the name of the instrument in the Methods.

REVIEWERS' COMMENTS

Reviewer #3 (Remarks to the Author):

During revisions the authors have robustly addressed all comments made in response to the initial submission. The revised manuscript makes an important and perhaps surprising contribution to skeletal biology through demonstration of the distinct ultimate fates of Fgfr3-lineage and Dlx5-lineage cells, and the conclusions are well justified by the data. There is a high level of enthusiasm for this manuscript.